# Direct then Diffuse: Incremental Unsupervised Skill Discovery for State Covering and Goal Reaching

## Abstract

Learning meaningful behaviors in the absence of a task-specific reward function is a challenging problem in reinforcement learning. A desirable unsupervised objective is to learn a set of diverse skills that provide a thorough coverage of the state space while being directed, i.e., reliably reaching distinct regions of the environment. At test time, an agent could then leverage these skills to solve sparse reward problems by performing efficient exploration and finding an effective goal-directed policy with little-to-no additional learning. Unfortunately, it is challenging to learn skills with such properties, as diffusing (e.g., stochastic policies performing good coverage) skills are not reliable in targeting specific states, whereas directed (e.g., goal-based policies) skills provide limited coverage. In this paper, inspired by the mutual information framework, we propose a novel algorithm designed to maximize coverage while ensuring a constraint on the directedness of each skill. In particular, we design skills with a decoupled policy structure, with a first part trained to be directed and a second diffusing part that ensures local coverage. Furthermore, we leverage the directedness constraint to adaptively add or remove skills as well as incrementally compose them along a tree that is grown to achieve a thorough coverage of the environment. We illustrate how our learned skills enables to efficiently solve sparse-reward downstream tasks in navigation and continuous control environments, where it compares favorably with existing baselines.

## 1 Introduction

Deep reinforcement learning (RL) algorithms have been shown to effectively solve a wide variety of complex problems [e.g., 23, 6, 31, 12, 2, 28]. However, they are often designed to solve one single task at a time and they need to restart the learning process from scratch for any new problem, even when it is defined on the very same environment (e.g., navigating to different locations in the same apartment). Recently, unsupervised RL (URL) has been proposed as an approach to address this limitation. In URL, the agent first interacts with the environment without any extrinsic reward signal. Afterward, the agent leverages the experience accumulated during the unsupervised learning phase to efficiently solve a variety of downstream tasks defined on the same environment.

In this paper, we consider the URL setting where the agent starts from an initial state $s_0$ and it resets to it every time the policy terminates. We focus on sparse-reward downstream tasks, which require effective exploration (i.e., via a thorough coverage of the state space) to find the goal as well as learning a policy reliably reaching the goal (i.e., a directed policy).

We build on the insight that *mutual information* (MI) effectively formalizes the dual objective of learning skills that both cover and navigate the environment efficiently [e.g., 11]. Specifically, given the state variable $S$ and some variables $Z$ on which the skill policies are conditioned, MI is defined as

$$\mathcal{I}(S;Z) = \underbrace{\mathcal{H}(S)}_{\text{coverage}} \underbrace{- \mathcal{H}(S|Z)}_{\text{directedness}} = \mathcal{H}(Z) - \mathcal{H}(Z|S), \tag{1}$$

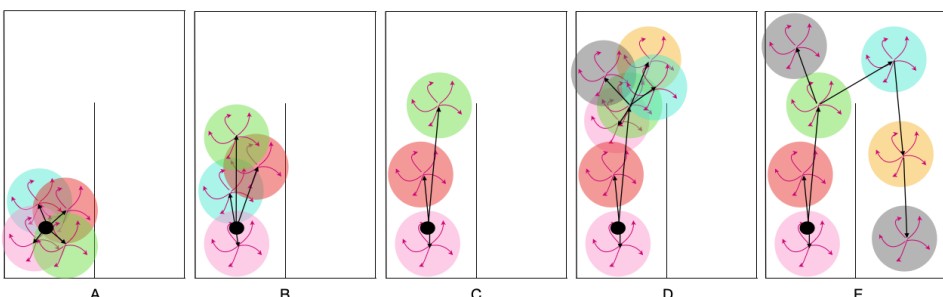

Figure 1: Overview of UPSIDE. The black dot corresponds to the initial state $s_0$. *(A)* A set of random skills is initialized, each skill being composed of a *directed* part (illustrated as a black arrow) and a *diffusing* part (red arrows), which induces a local coverage (colored circles). *(B)* The policies associated to the directed part of each skill are then updated to maximize the discriminability of the states reached by their diffusing part (Sect. 3.1). *(C)* The least discriminable skills are iteratively removed while the policies of the remaining skills are re-optimized. This is executed until the discriminability of each skill satisfies a given constraint (see Sect. 3.2). In this example three skills are kept. *(D)* One of these learned skill is then used as basis to add new skills, which are then optimized following the same procedure. For the "red" and "purple" skills, UPSIDE is not able to find sub-skills of sufficient quality and thus they are not expanded any further. *(E)* At the end of the process, UPSIDE has created a tree of directed skills covering the state space (Sect. 3.3). These covering skills can then be used to solve downstream tasks. Moreover, the discriminator learned together with the skills can be used to select the skill to reach any specific goal region, where the directed parts get close to the goal, while the diffusing part provides the local coverage to attain the goal. The complete algorithm is detailed in Sect. 3.4 and Appendix.

where $\mathcal{I}$ denotes the MI and $\mathcal{H}$ is the entropy function. The first expression, known as the forward form of MI, explicitly balances the two sought-after properties of *coverage* — captured by the entropy over the state space $\mathcal{H}(S)$ — and *directedness*, i.e., the ability to reach specific states $S$ depending on $Z$ — captured by the negative conditional entropy $-\mathcal{H}(S|Z)$. The second expression of (1), often easier to optimize and referred to as the reverse form, stipulates that the skills should be sampled as diversely as possible while being discriminable.

Maximizing (1) has been shown to be a powerful approach for encouraging exploration in RL [16, 25] and for unsupervised skill discovery [e.g., 11, 9, 1, 30, 8]. Nonetheless, learning skills that maximize the MI is a challenging optimization problem. Several approximations have been proposed to simplify the problem at the cost of possibly deviating from the original objective of coverage and directedness (see Sect. 4 for a review of related work). In this paper, we propose UPSIDE (*UnsuPervised Skills that dIrect then DiffusE*) to learn skills that can be effectively used to solve goal-based downstream tasks. Our solution builds on the following components (see Fig. 1 for an illustration of UPSIDE):

- *Skill structure.* In order to balance coverage and directedness, we design skills composed of two parts: **1)** a *directed* part that is trained to reach a distinct region of the environment, and **2)** a *diffusing* part that covers the states around the region attained by the first part.
- *Optimization.* We further strengthen the coverage and directedness properties of the skills by turning the MI objective into a constrained optimization problem designed to maximize coverage under the constraint that *each* skill achieves a minimum level of discriminability. This in turn enables UPSIDE to adaptively add skills to improve coverage, when all the initial skills meet the constraint, or remove those that violate the constraint to guarantee that each skill is directed and reaches a distinct region of the environment.
- *Tree structure.* When the agent starts from a fixed initial state, the skills' length is a crucial parameter, where short skills do not allow for proper coverage, and long skills are difficult to train. In UPSIDE we consider short skills to make the optimization easier, while composing them along a tree structure that ensures an adaptive and deep coverage of the environment.

We study how our learned skill structure enables to both perform efficient exploration and learn effective goal-reaching policies in a variety of navigation and continuous control environments (including MuJoCo's reacher) and we compare its performance to relevant baselines.

## 2  Setting

We consider the URL setting where the agent interacts with a Markov decision process (MDP) $M$ with state space $\mathcal{S}$, action space $\mathcal{A}$, dynamics $p(s'|s,a)$, and **no reward**. The agent starts each

episode from a designated initial state $s_0 \in \mathcal{S}$. Upon termination of the chosen policy, the agent is then reset to $s_0$. This setting is particularly challenging from an exploration point of view since the agent cannot rely on the initial distribution to cover the state space.

We recall the MI-based unsupervised skill discovery approach [see e.g., 11]. Denote by $Z$ some (latent) variables on which the skills of length $T$ are conditioned. There are three optimization variables: *(i)* the support of the skills denoted by $|Z|$ (we consider it to be discrete so $|Z|$ is the number of skills), *(ii)* the policy $\pi(z)$ associated to skill $z$, and *(iii)* the sampling rule $\rho$ (i.e., $\rho(z)$ is the probability of sampling skill $z$ at the beginning of the episode). Let the variable $S_T$ be the random (final) state induced by sampling a skill $z$ from $\rho$ and executing the associated policy $\pi(z)$ from $s_0$ for an episode. We denote by $p_{\pi(z)}(s_T)$ the distribution over (final) states induced by executing the policy of skill $z$, by $p(z|s_T)$ the probability of $z$ being the skill to induce state $s_T$, and let $\overline{p}(s_T) = \sum_{z \in Z} \rho(z) p_{\pi(z)}(s_T)$. Then maximizing the MI between $Z$ and $S_T$ can be written as

$$\max_{|Z|, \rho, \pi} \mathcal{I}(S_T; Z) = \mathcal{H}(S_T) - \mathcal{H}(S_T|Z) = -\sum_{s_T} \overline{p}(s_T) \log \overline{p}(s_T) + \sum_{z \in Z} \rho(z) \mathbb{E}_{s_T} \left[ \log p_{\pi(z)}(s_T) \right]$$

$$= \mathcal{H}(Z) - \mathcal{H}(Z|S_T) = -\sum_{z \in Z} \rho(z) \log \rho(z) + \sum_{z \in Z} \rho(z) \mathbb{E}_{s_T} \left[ \log p(z|s_T) \right], \quad (1)$$

where in the expectations $s_T \sim p_{\pi(z)}(s_T)$. As discussed in Sect. 1, learning the optimal $|Z|$, $\rho$, and $\pi$ is a challenging problem [see e.g., 11, 9, 8].

## 3 Algorithm Structure

UPSIDE is based on three main components: **a)** the skill learning corresponding to stage $A$ and $B$ of Fig. 1 and described in Sect. 3.1, **b)** a constrained optimization problem used to optimize the number of skills (stage $C$ and Sect. 3.2) and **c)** a tree-building procedure (stage $D$ and Sect. 3.3). Together, these components allow UPSIDE to discover skills that combine coverage and directedness.

### 3.1 Skill Structure and Optimization

As shown in e.g., [9, 30, 37], the level of stochasticity of each skill (e.g., induced via a regularization on the entropy over the actions) plays a key role in trading off coverage and directedness. In fact, while randomness promotes broader coverage, it may compromise the directedness of the skills. In fact, a highly stochastic skill tends to induce a distribution $p_{\pi(z)}(s_T)$ over final states with high entropy (thus decreasing $-\mathcal{H}(S_T|Z)$), which prevents the skill to be reusable in solving sparse-reward downstream tasks where the objective is to reliably reach a specific goal state of the environment. Determining *how much* stochasticity to inject to adequately balance both objectives and optimize (1) is a difficult problem.[1]

We propose to design skills with a *decoupled policy structure*:

- A *directed* part (of length $T$) with low stochasticity and trained to reach a specific region of the environment. It is responsible for increasing the $-\mathcal{H}(S|Z)$ term in (1).

- A *diffusing* part (of length $H$) with high stochasticity to promote local coverage of the states around the region reached by the directed part. It is responsible for increasing the $\mathcal{H}(S)$ term in (1).

Figure 2: Directed and diffusing parts of the skill.

Similar to prior work [e.g., 11, 9], the policy associated to the directed part of skill $z$ is trained to maximize an intrinsic reward $r_z(s) \approx p(z|s)$,[2] where $p(z|s)$ measures the "discriminability" of the skill $z$ given the state $s$. More formally, $\pi(z)$ maximizes the cumulative reward $\mathbb{E}_{\pi(z)} \left[ \sum_{t=T+1}^{T+H} r_z(s_t) \right]$ over the states traversed by the policy during the diffusing part. In practice, we also add a small entropy regularization $\mathcal{H}(\pi(\cdot|z, s_t))$ to the directed policy in order to ensure a minimum level of exploration and make the learning more robust. For the diffusing part, we rely on a simple random walk policy (i.e., a stochastic policy with uniform distribution over actions).

---

[1]In RL, stochasticity is injected at "train time" to boost *exploration* or improve *robustness*, while the policy executed at "test time" is deterministic. Here we refer to stochasticity introduced to better optimize (1).

[2]Although [11, 9] employ rewards in the log domain, we find that using a reward that is a non-linear transformation into $[0, 1]$ works better in practice, as also observed in [33, 5]. Furthermore, in practice we replace $p(z|s)$ by the predictions of a learned discriminator $q_\phi(z|s)$ as explained in Sect. 3.4.

Intuitively, the diffusing part defines a cluster of states that is used as a goal for the directed part. This allows us to "ground" the latent variable representations of the skills $Z$ to specific regions of the environment (i.e., the clusters). As a result, maximizing the MI over such skills can be seen as learning a set of "cluster-conditioned", and thus directed, policies.

## 3.2 Skill Support and Sampling Rule

The MI objective (1) crucially depends on the number of skills ($|Z|$) and the distribution $\rho(z)$. Unfortunately, it is been shown [e.g., 8] that solving (1) is particularly challenging. In order to simplify the optimization and the associated learning problem, we modify (1) in two ways.

First, coherently with the skill optimization detailed in Sect. 3.1, the random variable $S$ in the conditional entropy is any state reached during the diffusing part of the skill and not just the terminal state. More formally, we denote by $S_{\text{diff}}$ the random variable and its distribution for a specific skill $z$ is $p_{\pi(z)}(s_{\text{diff}}) = 1/H \sum_{t=T+1}^{T+H} p_{\pi(z)}(s_t)$, i.e., the distribution over states obtained by averaging the distributions at any of the steps in the diffusing part. Similarly, $p(z|s_{\text{diff}})$ now denotes the probability of $z$ being the skill to traverse $s_{\text{diff}}$ during its diffusing part. As a result, training the skills to maximize MI naturally leads the diffusing parts to "push" the directed parts away so as to reach diverse regions of the environment. The combination of "global" coverage of the directed parts and "local" coverage of the diffusing part ensures that the whole environment is properly visited with $|Z| \ll S$ skills.[3]

Second, we introduce an alternative problem that simplifies the optimization while preserving the coverage and directedness properties of MI. This is achieved by introducing a stronger requirement on the discriminability. While the conditional entropy term $-\mathcal{H}(Z|S)$ in (1) promotes the discriminability of skills *on average*, we argue that a more suitable objective is to *constrain* each skill to achieve a *minimum* level of discriminability. First, we move from the average to the minimum over skills by lower bounding the conditional entropy as

$$-\mathcal{H}(Z|S_{\text{diff}}) = \sum_{z \in Z} \rho(z) \mathbb{E}_{s_{\text{diff}}} \left[ \log p(z|s_{\text{diff}}) \right] \geq \min_{z \in Z} \mathbb{E}_{s_{\text{diff}}} \left[ \log p(z|s_{\text{diff}}) \right], \tag{2}$$

which leads to the following optimization (assuming $\pi$ is fixed for convenience)

$$\max_{|Z|=N,\rho} \left\{ \mathcal{H}(Z) + \min_{z \in [N]} \mathbb{E}_{s_{\text{diff}}} \left[ \log p(z|s_{\text{diff}}) \right] \right\}, \tag{3}$$

where with an abuse of notation we use $z \in [N]$ to denote all skills in a set $Z$ with cardinality $N$. Since (3) is a lower bound to MI, it tends to promote the same type of covering and directed skills. Furthermore, (2) no longer depends on the distribution over skills and the entropy term $\mathcal{H}(Z)$ is maximized by setting $\rho$ to the uniform distribution over $N$ skills (i.e., $\max_\rho \mathcal{H}(Z) = \log(N)$), thus simplifying the optimization, which now only depends on $N$.

While optimizing (3) promotes a cardinality $N$ such that all skills have good discriminability, a more convenient formulation is to explicitly set a minimum level of discriminability for all skills through the following constrained optimization problem:

$$\max_{N \geq 1} \log(N) \qquad \text{s.t.} \qquad \min_{z \in [N]} \mathbb{E}_{s_{\text{diff}}} \left[ \log p(z|s_{\text{diff}}) \right] \geq \log \eta. \tag{4}$$

where $\eta$ is a parameter that defines the discriminability threshold. A skill $z$ is said to be $\eta$-*consolidated* if it satisfies the constraint. Crucially, let $P_N := \min_{z \in [N]} \mathbb{E}_{s_{\text{diff}}} \left[ \log p(z|s_{\text{diff}}) \right]$, then the sequence $(P_N)_{N \geq 1}$ is non-increasing with $P_1 = 0$ (i.e., the more skills the harder it is to meet the constraint). As a result, (4) can be optimized following a simple greedy strategy incrementally adding skills until the constraint is violated. The optimal $N$ thus defines the *effective number* of $\eta$-consolidated skills and it corresponds to the largest number of skills that is guaranteed to display sufficient discriminability. Alternatively, we can interpret (4) as finding the largest number of clusters (i.e., the region reached by the directed part of a skill and covered by its associated diffusing part) with a minimum level of inter-cluster distance. This effect is qualitatively illustrated in Fig. 1, where the states attained by the directed part of the skills attain different regions that are locally covered by their diffusing parts.

---

[3]Notice that (1) is maximized by setting $|Z| = |S|$ (since $\max_Y \mathcal{I}(X, Y) = \mathcal{I}(X, X) = \mathcal{H}(X)$), i.e., where each skills is a goal-conditioned policy reaching a different state. This implies having as many policies as states, which makes the learning particularly challenging as the complexity of the environment increases.

**Algorithm 1:** UPSIDE

---

**Initialize**: Discriminability threshold $\eta \in (0, 1)$, branching factor $N_0 \geq 1$, patience $K$
**Initialize**: Tree $\mathcal{T}$ initialized as a root node indexed by 0, queue of parent nodes $\mathcal{W} = \{0\}$.
**while** $\mathcal{W} \neq \emptyset$ **do** // `tree expansion`

  1    Dequeue a node/skill $w \in \mathcal{W}$ and expand $\mathcal{T}$ at $w$ by adding a set $\mathcal{C}(w)$ of $N_0$ nodes/skills
  2    Create random policies $\pi_z,\ \forall z \in \mathcal{C}(w)$
  3    Initialize discriminator $q_\phi$ with $|\mathcal{T}|$ classes
  4    `Continue = true; Saturated = false`
  5    **while** `Continue` **do**
  6       **for** $K$ iterations **do**
  7          Sample a skill $z$ from $\mathcal{T}$ at random
  8          Extract the sequence of nodes $z_{(1)}, \ldots, z$ in $\mathcal{T}$ leading to $z$
  9          Execute the composed (directed part) policy $(\pi_{z_{(1)}}, \ldots, \pi_z)$ followed by the diffusing part
 10          Add states observed during the diffusion part to state buffer $\mathcal{B}_z$
 11          Update discriminator $q_\phi$ with SGD on $\mathcal{B}_z$ to predict label $z$
 12          **if** $z \in \mathcal{C}(w)$ **then** // `Update only new policies, other polices kept fixed`
 13             Update policy $\pi_z$ using SAC to optimize the discriminator reward as in Sect. 3.1.
 14       Compute the skill-discriminability $d(z) = \widehat{q}_\phi^{(B)}(z) = \frac{1}{|\mathcal{B}_z|} \sum_{s \in \mathcal{B}_z} q_\phi(z|s)$ for all $z \in \mathcal{C}(w)$
 15       **if** $\min_{z \in \mathcal{C}(w)} d(z) < \eta$ **then** // `Node removal`
 16          Remove the node/skill $z = \arg\min_{z \in \mathcal{C}(w)} d(z)$ from $\mathcal{C}(w)$ and $\mathcal{T}$
 17          Set `Saturate = true`
 18       **else if** `not Saturated` **then**
 19          Add one new node/skill to $\mathcal{C}(w)$ and $\mathcal{T}$
 20       **else**
 21          Set `Continue = false`
 22    Enqueue in $\mathcal{W}$ the consolidated nodes $\mathcal{C}(w)$

---

## 3.3 Composing Skills in a Tree Structure

The MI optimization problem as well as our constrained variant (4) depend on the initial state $s_0$ and on the length of each skill. Although these quantities are usually predefined and only appear implicitly in the equations, they have a crucial impact on the obtained behavior. In fact, resetting after each skill execution unavoidably restricts the coverage to a radius of at most $T + H$ steps around $s_0$. This may suggest to set $T$ and $H$ to a large value. However, increasing the horizon makes the training of the skills more challenging, as learning $\pi$ would require solving a difficult RL problem itself.

Instead, we propose to "extend" the length of the skills through composition. Indeed, the decoupled skill structure and the constraint in (4) entail that the directed part of each of the $\eta$-consolidated skills reliably reach a specific (and distinct) region of the environment and it is thus re-usable and amenable to composition. We propose to chain the directed part of the skills in order to reach further and further parts of the state space. Specifically, we build a growing tree, where the root is the initial state $s_0$, the edges represent the directed part of the skills, and the nodes represent the diffusing part of skills. As such, whenever a skill $z$ is selected, the directed part of all the policies associated to its predecessor skills in the tree are executed first (see Fig. 1 for an illustration of the tree structure).

As a result, the agent naturally builds a curriculum on the episode lengths, which grow as the sequence $(iT + H)_{i \geq 1}$. As such, it does not require prior knowledge on an adequate horizon of the downstream goal-based task.[4] Here this knowledge is replaced by $T$ and $H$ which are more environment-agnostic and task-agnostic quantities, as their choice rather has an impact on the size and shape of the learned tree (e.g., the smaller $T$ and $H$ the bigger the tree).

## 3.4 The UPSIDE Algorithm

We are now ready to introduce UPSIDE, which provides a specific implementation of the components described before (see Fig. 1 for a qualitative illustration and Algorithm 1 for the detailed pseudo-code).

We perform standard approximations to make the constraint in (4) easier to estimate. We approximate the unknown posterior $p(z|s)$ with a learned discriminator $q_\phi(z|s)$ with parameters $\phi$. We also

---

[4]See e.g., the discussion in [26] on the "importance of properly choosing the training horizon in accordance with the downstream-task horizon the policy will eventually face."

remove the logarithm from the constraint to have an estimation range of $[0, 1]$ and thus lower variance[2]. Finally, we replace the expectation over $s$ with an empirical estimate $\widehat{q}_\phi^{(B)}(z)$ averaging the value of the discriminator evaluated on the last $B$ states observed while executing the diffusing part of $z$. Integrating these approximations in (4) leads to

$$\max_{N \geq 1, \pi} N \qquad \text{s.t.} \qquad \min_{z \in [N]} \widehat{q}_\phi^{(B)}(z) \geq \eta. \tag{5}$$

As discussed in Sect. 3.2, this problem can be conveniently optimized using a greedy strategy. We then integrate the optimization of (5) into an adaptive tree expansion strategy: **(Generating new skills)** Given a tree structure as described in Sect. 3.3, we expand the tree at a leaf $w$ by adding $N_0$ new nodes/skills following a breadth-first-search approach (lines 1, 2). Then **(Skill Learning)** the new skills are optimized by: **i)** sampling random skills in the tree to update the discriminator (lines 7-11), and **ii)** by updating the policies to optimize the discriminability reward (Sect. 3.1) computed using the discriminator (lines 13). To speed-up convergence, we only update the policies that have be added to the tree structure, keeping all the previous policies fixed (line 12). Note that in the update of the discriminator we leverage the states observed in previous phases of the algorithm by maintaining a (small) replay buffer of states for each skill. **(Node Consolidation)** After a *patience* period (line 6), if all skills are $\eta$-consolidated, we tentatively add more skills to the leaf $w$ (line 18). On the other hand, if any skill does not meet the discriminability threshold, we remove it and consolidate the remaining skills into the tree (lines 16, 17) and we repeat the process.

**Model selection.** A core aspect of any RL algorithm is *model selection*, i.e., finding the best configuration of hyperparameters. In URL with no prior knowledge of the downstream task(s), it is non-trivial to devise an adequate criterion for model selection and this aspect is rarely addressed, despite being crucial in practice. For instance, while the coverage of the state space may be a good proxy for the performance of a URL algorithm [see e.g., 8], it may be difficult to measure in continuous problems. Interestingly, our optimization problem directly provides a single, task-agnostic and environment-agnostic criterion for model selection, which is the number $N$ of $\eta$-consolidated skills discovered by the agent. Indeed in all of our experiments we simply select the model (i.e., set of hyperparameters) that maximizes $N$. This is a significant advantage w.r.t. existing methods, such as VIC and DIAYN, for which no principled approach to model selection is provided.

# 4 Related work

Unsupervised Reinforcement Learning methods can be broadly decomposed according to the way they summarize the experience accumulated during the unsupervised phase into reusable knowledge to solve downstream tasks. This includes both off-policy model-free [e.g., 27] and model-based [e.g., 29] methods that seek to populate a representative replay buffer and build accurate value or model estimates, that are used to solve a given downstream task in a zero- or few-shot manner. The accumulated experience during train time can also be compressed into a low-dimensional representation for value functions as well as policies and to improve exploration [e.g., 36]. An alternative line of work focuses on the discovery of a set of skills in an unsupervised manner. Our approach falls in this category, on which we now focus our related work review.

Skill discovery based on MI maximization was first proposed in VIC [11], where only the final states of each trajectory are considered in the reverse form of (1) and where both the skills and their sampling rules are simultaneously learned (with a fixed support $|Z|$, i.e., a fixed number of skills). DIAYN [9] fixes the sampling rule to be uniform, and weighs the skills with an action-entropy coefficient (i.e., it additionally minimizes the MI between actions and skills given the state), so as to push the skills away from each other and enhance coverage. DADS [30] learns skills that are not only diverse but also predictable by learned dynamics models, by using a generative model over observations (rather than over skills) and optimizing a forward form of MI, namely $\mathcal{I}(s'; z|s)$ between the next state $s'$ and current skill $z$ (with continuous latent) conditioned on the current state $s$. EDL [8] shows that existing skill discovery approaches can provide insufficient coverage, and instead proposes to rely on a fixed distribution over states $p(s)$ which is either provided by an oracle or learned. In SMM [19], the MI formalism is used to learn a policy for which the state marginal distribution matches a given target state distribution (e.g., uniform), which can be seen as a more scalable way of tackling the problem of maximum entropy over the state space [15], and as a way to encourage skills to go through unknown state regions. Other MI-based skill discovery methods include [10, 14, 24, 5, 34], as well as [35, 20] which investigate skill discovery in non-episodic settings.

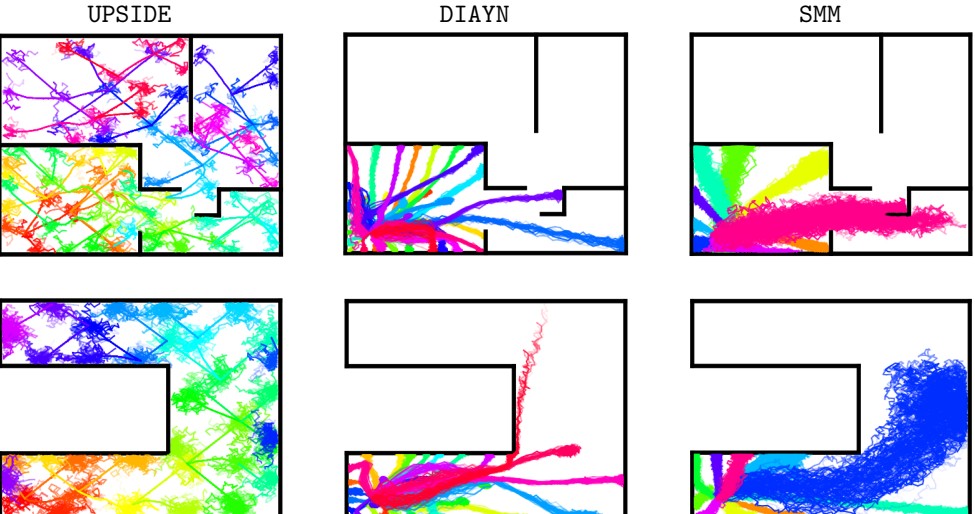

Figure 3: UPSIDE, DIAYN-curriculum and SMM-10 skills learned in a bottleneck maze *(Top)* and a U-maze *(Bottom)*. For both DIAYN and SMM we report the stochastic execution of the learned skills and for UPSIDE we report the deterministic directed parts (that are composed) followed by the (stochastic) diffusing part, which is the same protocol used to evaluate coverage.

Our approach shares a similar motivation to prior MI-based works of targeting skills that are both directed and state-covering. In particular, the decoupled structure introduced in Sect. 3.1 can be seen as a more suitable way to achieve the objective of improving the coverage of VIC as done in DIAYN and SMM, without compromising the directedness of the skills.

While most skill discovery approaches consider a fixed number of skills, a curriculum with increasing number of skills is studied in [1, 3]. Our discriminability constraint is what enables skills to be composed along a tree structure, which allows increases or decreases the support of available skills depending on the region of the state space.

Recently, [37] proposed a hierarchical RL method that discovers abstract and task-agnostic skills while jointly learning a higher-level policy which is trained to maximize environment reward. Our approach builds on a similar promise of composing skills instead of resetting to $s_0$ after each execution, yet we articulate the composition differently, by exploiting the direct-then-diffuse structure to ground learned skills to the state space instead of being abstract.

In addition, approaches such as DISCERN [33] and Skew-Fit [27] learn a goal-conditioned policy in an unsupervised way with an MI objective. As explained in [8, Sect. 5], this can be interpreted as a skill discovery approach with latent $Z = S$, i.e., where each goal state can define a different skill. Conditioning on either goal states or abstract latent skills forms two extremes of the spectrum of unsupervised RL. We target an intermediate approach, seeking to benefit from the groundedness of the latent skill $Z$ and the states $S$ (and thus amenability to composition) of goal-conditioned RL, and from the reduced search space and sampling ease of skill-based RL.

An alternative approach to skill discovery builds on "spectral" properties of the dynamics of the environment. This includes eigenoptions [21, 22] and covering options [17, 18], as well as the algorithm of [4] that builds a discrete graph representation which learns and composes spectral skills.

## 5  Experiments

In this section, we investigate the following questions: **i)** Can the adaptive tree structure of UPSIDE incrementally cover an unknown environment while preserving directedness of the skills? **ii)** Following the unsupervised phase, how can UPSIDE be leveraged to solve goal-based downstream tasks?

We report results on: **a)** Navigation problems in continuous mazes, where actions represent the desired shift in $x$ and $y$ coordinates; **b)** A difficult instance of CartPole, where the cart starts with zero speed and the pole is oriented downside; **c)** The Reacher [32] problem using the MuJoCo implementation in Gym [7]. In all environments, the per-dimension action space is in $[-1; +1]$.

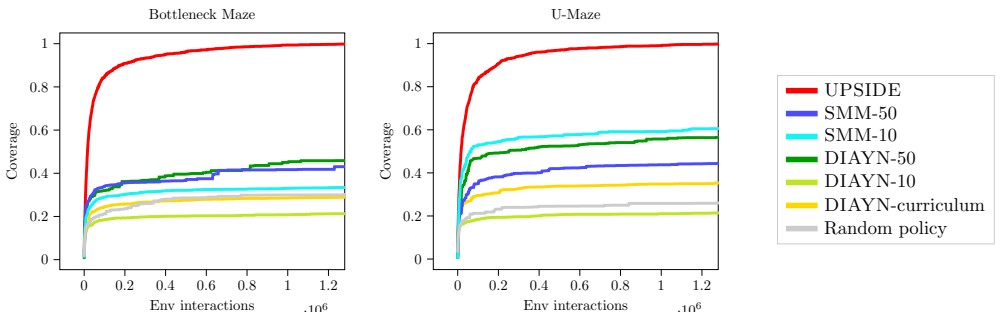

Figure 4: Normalized coverage in U-maze and bottleneck.

We compare to different baselines. DIAYN-K, where $K$ is a fixed number of skills, is the original algorithm proposed in [9]. DIAYN-Curriculum is a variant where the number of skills is automatically tuned following the same procedure as in UPSIDE ensuring a good discriminability. We also compare to SMM [19], which is similar to DIAYN, but it includes an exploration bonus encouraging the policies to visit rarely encountered states. In our implementation, the exploration bonus is obtained by maintaining a multinomial distribution over "buckets of states" obtained by discretization, resulting in an computation-efficient and stable implementation that is more stable than the original VAE-based method. UPSIDE and all baselines are implemented with Soft-Actor Critic (SAC) [13].

**Unsupervised Phase.** We run all methods until convergence. We then do model selection according to the criterion of either the final number of skills for UPSIDE and DIAYN-curriculum and the final average discriminability for DIAYN-K and SMM. To compute the coverage, we perform rollouts by first sampling a skill uniformly at random and executing its associated policy until termination. We discretize states into buckets (50 interval per dimension for mazes and 10 for control environments) and report the proportion of buckets reached by each method as a function of the total number of steps executed in the environment over multiple rollouts. Since only a small portion of the discretized states can be reached, we normalize the coverage such that the best method obtains 1.

We consider two topologies of mazes with size (height and width) 50 such that exploration is non-trivial (i.e., a random policy is only able to cover a small part of the state space): a U-shaped maze and a Bottleneck maze (which is a harder version of the one in [8, Fig. 1] which is only of size 10 for the same action space). In Fig. 3 we show that UPSIDE succeeds in covering the near-entirety of the state space by creating a tree of directed skills. Moreover, UPSIDE created directed skills with a low entropy, while the two baselines tend to create skills that are more stochastic. This is particularly evident for SMM, due to the state-entropy exploration bonus, that while it encourages broader coverage makes skills less directed.

In Fig. 4 we report the coverage on the Bottleneck maze and U-Maze. For UPSIDE, executing a skill corresponds to executing the directed part of all the "parent" skills in the tree and concluding with the diffusion part of the skill. SMM achieves better coverage than DIAYN thanks to the increased level of stochasticity (diffusion) of its skills. UPSIDE outperforms both by reaching regions of the environment that are not be achieved by other methods. Here, we plot UPSIDE with $T = 10$ and $H = 10$, but we found UPSIDE to be robust to these parameters as shown in the supplementary.

Results are similar in the CartPole problem (see Fig. 5) where UPSIDE (with $T = 20$ and $H = 20$) obtains better coverage than baselines. On the other hand, in Reacher (see Fig. 5), DIAYN-50 outperforms UPSIDE in terms of coverage. This can be explained by the fact that, in this environment, highly stochastic skills provide a good coverage. Nonetheless, this comes at the cost of very low discriminability (rightmost plot), which suggests DIAYN-50 skills have poor directedness. On the other hand, UPSIDE (and DIAYN-curriculum) achieves much larger discriminability by removing redundant skills and favoring more directed policies.

**Downstream Tasks.** Following the unsupervised phase, UPSIDE has learned a tree of skills. We now investigate how these skills are used to tackle a downstream task. In that setting, we propose to use skill-based approaches (i.e UPSIDE, DIAYN and SMM) in the following way: a) (exploration) first we sample rollouts over the different skills. b) We then select the best skill based on the maximum cumulative reward collected and c) we fine-tune this skill to maximize the reward. We report results on mazes (additional results are provided in the supplementary). We consider a sparse positive reward

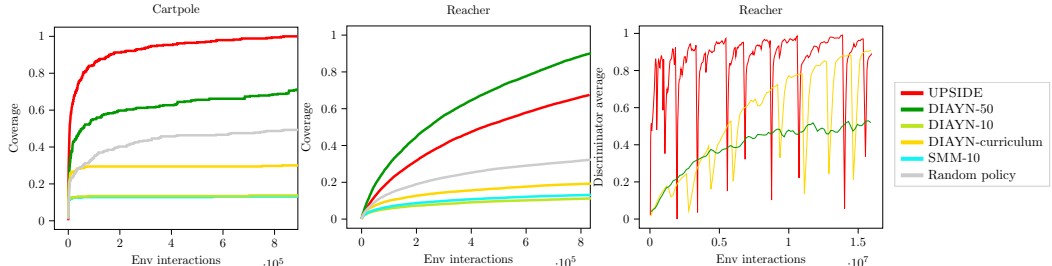

Figure 5: Normalized coverage in Cartpole *(Left)* and Reacher *(Middle)*. *(Right)* Average discriminability of the skills during training in Reacher.

when reaching a particular defined goal.[5] We consider goals at different distances from the initial state $s_0$, the further, the harder. Fig. 6 shows the learning curves obtained when fine-tuning the best skill for the different models and compare to a classical SAC algorithm where a single policy is learned from scratch. DIAYN/SMM means we use the best state-covering policies between DIAYN and SMM. For the "close" goal setting, both UPSIDE and DIAYN/SMM are able to learn to reach this goal efficiently while SAC solves the task only for some of the training runs. Note that we do not show DIAYN performance since it is lower than the SMM one. For the "far" goal setting, only UPSIDE learns to reach this goal. Obtained trajectories are illustrated in Fig. 6.

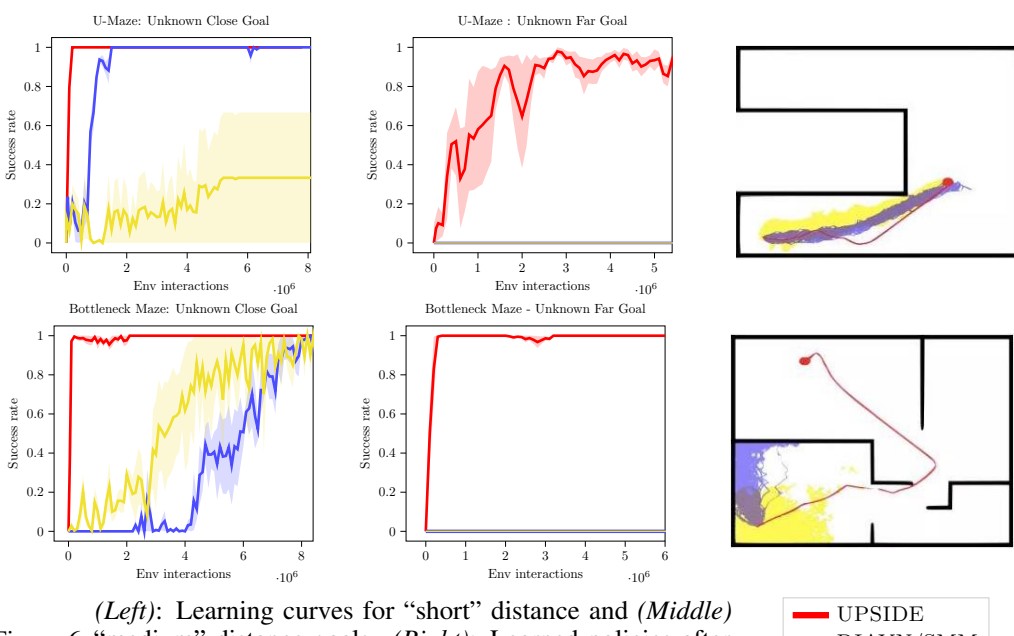

Figure 6: *(Left)*: Learning curves for "short" distance and *(Middle)* "medium" distance goals. *(Right)*: Learned policies after fine-tuning *(Top)* U-maze. *(Bottom)*: Bottleneck maze.

## 6 Conclusion

We introduced UPSIDE, a novel algorithm for unsupervised skill discovery designed to trade off between coverage and directedness and develop a tree of skills that can be used to both perform efficient exploration of the environment and learn effective goal-directed policies. Natural venues for future investigation are: **1)** The diffusing part of each skill could be explicitly trained to maximize local coverage; **2)** UPSIDE assumes a good representation of the state is provided as input, it would be interesting to pair UPSIDE with effective representation learning techniques to tackle problems with high-dimensional input (e.g., image-based RL); **3)** While UPSIDE is grounded on the solid principle of MI maximization, a more thorough theoretical investigation is needed to explicitly link the optimization problem and its approximations to the downstream performance.

---

[5]Notice that if the goal was known, the learned discriminator could be directly used to identify the most promising skill to fine-tune.

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
