# Appendix

## Table of Contents

## A   `UPSIDE` **Algorithm**

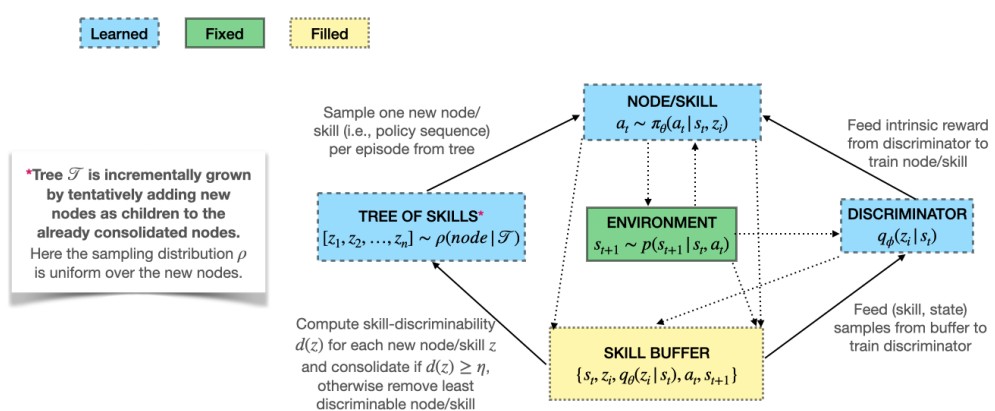

Figure 7: High-level approach of `UPSIDE`

We provide a diagram of the high-level approach of `UPSIDE` in Fig. 7 and a detailed pseudo-code in Alg. 2. `UPSIDE` initializes a tree structure $\mathcal{T}$ with root node 0 and queue of parent nodes $\mathcal{W} = \{0\}$. As long as the queue is not empty, the following steps are performed:

- **(Generating new skills)** We expand the tree at a leaf $w \in \mathcal{W}$ by adding $N_0$ new nodes/skills denoted by $\mathcal{C}(w)$ (lines 1, 2). We initialize a discriminator with $|\mathcal{T}|$ classes to account for the newly created nodes (line 3).

- (**Skill Learning**) Then the new skills and discriminator are optimized as follows:

   - We sample (uniformly) and rollout the new skills $z \in \mathcal{C}(w)$ and add the states of their diffusing parts in their corresponding buffers $\mathcal{B}_z$ (lines 8 to 11).

   - We update the discriminator by leveraging the states and skill labels in the buffers (lines 12 to 17). In particular, the ratio $\mu$ signifies that we train more often the discriminator on the previously consolidated skills/classes than on the new skills/classes in $\mathcal{C}(w)$, i.e., we sample pairs (label $z$, state in $\mathcal{B}_z$) with probability $(\mathbb{I}[z \in |\mathcal{C}(w)|] + \mu \mathbb{I}[z \notin |\mathcal{C}(w)|])/((1 - \mu)|\mathcal{C}(w)| + \mu|\mathcal{T}|)$. We give slightly more weight to the already consolidated skills in the discriminator training because the

**Algorithm 2:** UPSIDE

**Initialize**: Discriminability threshold $\eta \in (0,1)$, branching factor $N_0 \geq 1$ (to be adapted at each node), (optional) maximum branching factor $N_{\max} \geq N_0$, patience $K$, window $\mathcal{V}$, number of trajectory rollouts $R$ per update of discriminator and policies, batch size of $N_{\mathrm{discr}}$ to train the discriminator, ratio $\mu$ of probabilities between consolidated classes and new classes to train discriminator

**Initialize**: Tree $\mathcal{T}$ initialized as a root node indexed by 0, queue of parent nodes $\mathcal{W} = \{0\}$.

**while** $\mathcal{W} \neq \emptyset$ **do** // tree expansion

1    Dequeue a node/skill $w \in \mathcal{W}$ and expand $\mathcal{T}$ at $w$ by adding a set $\mathcal{C}(w)$ of $N_0$ nodes/skills

2    Create random policies $\pi_z$ and buffers $\mathcal{B}_z$, $\forall z \in \mathcal{C}(w)$

3    Initialize discriminator $q_\phi$ with $|\mathcal{T}|$ classes

4    Continue = true; Saturated = false

5    **while** Continue **do**

6      **for** $K$ iterations **do**

7        **for** $r \in [\![1, R]\!]$ **do** // collect $R$ trajectories

8          Sample a skill $z$ from $\mathcal{C}(w)$ at random

9          Extract the sequence of nodes $z_{(1)}, \ldots, z$ in $\mathcal{T}$ leading to $z$

10          Execute the composed (directed part) policy $(\pi_{z_{(1)}}, \ldots, \pi_z)$ followed by the diffusing part

11          Add states observed during the diffusing part to $\mathcal{B}_z$

12        $B = \{\}$ // Initialize batch to update the discriminator

13        **while** $|B| < N_{discr}$ **do**

14          Sample a skill $z$ from $\mathcal{T}$ w.p. $(\mathbb{I}[z \in |\mathcal{C}(w)|] + \mu\mathbb{I}[z \notin |\mathcal{C}(w)|])/((1-\mu)|\mathcal{C}(w)| + \mu|\mathcal{T}|)$

15          Sample a state $s$ from the last $\mathcal{V}$ states of $\mathcal{B}_z$

16          Add $(s, z)$ to $B$

17        Update discriminator $q_\phi$ with SGD on $B$ to predict label $z$

18        **for** $z \in \mathcal{C}(w)$ **do**

19          Update policy $\pi_z$ using SAC to optimize the discriminator reward as in Sect. 3.1.

20      Compute the skill-discriminability $d(z) = \hat{q}_\phi^{(B)}(z) = \frac{1}{|B|}\sum_{(s,z)\in B} q_\phi(z|s)$ for all $z \in \mathcal{C}(w)$

21      **if** $\min_{z \in \mathcal{C}(w)} d(z) < \eta$ **then** // Node removal

22        Remove the node/skill $z = \arg\min_{z \in \mathcal{C}(w)} d(z)$ from $\mathcal{C}(w)$ and $\mathcal{T}$

23        Set Saturate = true

24      **else if** not Saturated **then**

25        Add one new node/skill to $\mathcal{C}(w)$ and $\mathcal{T}$

26        **if** $|\mathcal{C}(w)| = N_{\max}$ **then**

27          Set Saturate = true

28      **else**

29        Set Continue = false

30    Enqueue in $\mathcal{W}$ the consolidated nodes $\mathcal{C}(w)$

---

discriminator is reinitialized whenever new classes (i.e., nodes) are added, thus we seek to avoid the new classes from invading the territory of the older classes that were previously correctly learned. In addition, we only update the discriminator on recent batches of data from the buffers via the window $\mathcal{V}$ (which considers only the last $\mathcal{V}$ states in each skill buffer), which is more sample efficient than doing the discriminator update in a fully on-policy manner (e.g., [12]), especially in our setting where the discriminator changes over training as new skill-nodes (i.e., classes) are added.

   - We update the policies of the new skills/nodes in $\mathcal{C}(w)$ with SAC to optimize the intrinsic reward of the discriminator predictions as explained in Sect. 3.1 (line 19). Note that we keep fixed the policies of the previously consolidated nodes/skills, which makes the learning of the tree more stable.

• **(Node Consolidation)** After a *patience* period characterized by $K$ iterations of training (line 6), if all skills are $\eta$-consolidated (i.e., the constraint of problem (5) is verified), we tentatively add more skills to the leaf $w$ (line 25). On the other hand, if any skill does not meet the discriminability threshold, we remove it and seek to consolidate the remaining skills into the tree (line 22). The role of the Saturated and Continue booleans is to ensure that the node addition operation cannot be performed if a node removal operation has already been performed in the training of the set $\mathcal{C}(w)$. Recall that the function is monotone, so if a skill is removed, the optimum cannot be larger. The (optional) $N_{\max}$ value represents the maximum branching factor (i.e., number of children nodes) imposed at each node of the tree.

## B  Environment Details

**Continuous mazes.**  We consider mazes with height and width 50. The state space is continuous, and there are some horizontal and verticals walls of width 1. The agent observes its current $(x, y)$ Cartesian position (i.e., it does not observe the walls) and it outputs actions $[dx, dy]$ that control its location. The actions $dx$ and $dy$ are constrained to be in $[-1, +1]$. The movement of the agent is affected by collisions with walls: when the agent collides with a wall, it stays in its original position.

**CartPole.**  We modify slightly the simulator from OpenAI Gym [8] to make exploration more difficult and thus to make it more challenging to learn diverse behaviors: the agent moves along the $x$ horizontal position between $-2.4$ and $2.4$ and the pole starts in the reverse position at $x = 0$. When the agent goes out of the $x$ interval, it is teleported back to its initial position (but there is no reset). Observations are $(x, \dot{x}, \theta, \dot{\theta})$ where $x$ is the horizontal position, $\theta$ is the angle of the pole to the $x$-axis, and $\dot{x}, \dot{\theta}$ are their respective velocities.

**Reacher.**  We use the standard MuJoCo implementation of Reacher [41], which is a two-joint robotic arm where the action space ($[-1, +1]$) is the torque applied to both joints with gear 30.

## C  Experimental Details

### C.1  Baselines

For all methods, we augment the state space with the current time-step because horizons are finite.

**DIAYN-K.**  This corresponds to the original `DIAYN` algorithm [12] where $K$ is the number of skills to be learned. In order to make the architecture more similar to `UPSIDE`, we use distinct policies for each skill, i.e. they do not share weights as opposed to [12]. While this may come at the price of sample efficiency, it may also help put lesser constraint on the model (e.g. gradient interference).

**DIAYN-Curriculum.**  We augment `DIAYN` with a curriculum that enables to be less dependant on an adequate tuning of the hyperparameter of the number of skills of `DIAYN`. We consider the curriculum of `UPSIDE` where we start from either a large or small number $N_0$ of skills, learn skills during a period of time/number of interactions. If the configuration satisfies the discriminablity threshold $\eta$, a skill is added, otherwise a skill is removed or learning stopped (as in Alg. A, lines 20-29). Note that the increasing version of this curriculum is similar to the one proposed in `VALOR` [1, Sect. 3.3].

**SMM.**  We used SMM [24] as it is state-of-art in terms of coverage, at least on long-horizon control problems, although [10] reported poor performance in hard-to-explore bottleneck mazes. We tested the regular SMM version, i.e. learning a state density model with a VAE, yet we failed to make it work on the Mazes domain. As we use the cartesian $xy$ positions in maze domains, learning the identity function on two-dimensional input data is too easy with a VAE, thus preventing the benefits of using a density model to drive exploration. Thus we considered a more straightforward implementation of SMM by using the "real" state distribution through counting. Specifically, we maintain a discretized state distribution by counting states in buckets (similar to the way we compute the achieved coverage). The distribution is just computed by dividing by the sum over buckets. We did not use a moving average so counts are not forgotten: the state distribution is over all policies encountered since the beginning of training (whereas the state distribution is "online" in [24]).

### C.2  Architecture and Hyperparameters

The architecture of the different methods remains the same in all our experiments, except that the number of hidden units changes across considered environments. We consider decoupled actor and critic in SAC, they both have the same (but unshared weights) state processing architectures. The observation and the step are passed through non-linear MLP with 1 hidden layer with units $h$, then are concatenated. The concatenation is then mapped to an embedding. For the actor, this embedding is mapped to a mean and variance embedding, then passes through a Squashed Gaussian as explained

in [17]. For the critic, the embedding is concatenated with a non-linear (1 hidden layer) embedding of the action, then passed through a final non-linear MLP (1 layer) to a one-dimensional value.

The discriminator is a two-hidden layer model with output size the number of skills in the tree.

**Environment-specific hyperparameters.** Mazes: $h = \{16, 64\}$ hidden units per layer for policy, and $h = 128$ hidden units per layer for discriminator. Continuous control domains: $h = 256$ hidden units per layer for both policy and discriminator.

**Common (for methods and environments) optimization hyperparameters.** (See App. A for meaning of each hyperparameter)

- SAC entropy: $\{0.1, 0.01, 0.001\}$
- discount factor: $\gamma = 0.99$
- Q-function soft updates $\tau = 0.005$
- learning rates $lr_{\text{policy}} = 0.001$, $lr_{\text{discriminator}} = \{0.0001, 0.001\}$
- discriminator batch size $B = 1024$
- $\mu = \{2, 5\}$
- $\mathcal{V} = 100$
- Replay buffer size: $1e6$

Note that hyperparameters are kept fixed for the downstream tasks too.

For UPSIDE and DIAYN-curriculum, we set the patience to be a time-limit instead of a number of iterations. We tried both 300 and 600 seconds to avoid the running time getting too high if the tree grows large.

The total running time for DIAYN-K and SMM is the same than the maximum running time of UPSIDE.

## C.3  Model selection

We train all methods with a grid search over the set of hyperparameters described in App. C.2, for multiple seeds, which we call *unsupervised seeds*, to evaluate robustness over both the initialization of model weights and randomness of the algorithm. For each unsupervised seed, we select the set of hyperparameters that has maximum value for the criterion of number of skills for UPSIDE, DIAYN-curriculum and for the criterion of average discriminability for DIAYN-K and SMM.

With this set of hyperparameters per seeds, we can then report some measurement, e.g. coverage, averaged over unsupervised seeds.

## C.4  Evaluation protocol

1. We train the method in its unsupervised phase.

2. We then do model selection as explained in App. C.3, which gives a model per method per unsupervised seed.

3. We rollout $N$ episodes per model and compute coverage as explained in the main paper in Sect. 5. Coverage is averaged over unsupervised seeds.

4. For each model (associated to a method) and unsupervised seed, we run the downstream tasks (as explained in App. C.5), with the same grid search over hyperparameters, with additional seeds, which we call *downstream seeds*.

5. For each method and unsupervised seed, we do model selection over downstream seeds on the criterion of reward.

6. We plot the reward averaged over unsupervised and downstream seeds, with error bars for each method.

## C.5 Downstream task scenario in Mazes

We consider the downstream task of quickly finding and then reliably reaching an unknown goal, summarized in Alg. 3. There exists a goal region $\mathcal{G}$ with unknown coordinates $(x_\mathcal{G}, y_\mathcal{G})$ that can be identified only once it is reached. The unknown nature of the goal and its sparse identification signal (i.e., reward $r_\mathcal{G}(s) = \mathbb{1}[s \in \mathcal{G}]$) makes the problem challenging, as the agent must perform "blind" and exhaustive exploration so as to encounter the goal as quickly as possible. UPSIDE's clustering of the state space with its ability to navigate efficiently to any given cluster is a desirable property to tackle this problem. In Alg. 3, we uniformly sample the nodes of the tree (i.e., execute the diffusing part of each skill) until the goal is found. Note that we use a budget of $K$ iterations (which could be either environment interactions or time) for UPSIDE to find the goal with the tree, otherwise we train a policy with SAC on the reward.

Once the goal is identified, this becomes a standard goal-oriented task, where no distance-to-goal is available, i.e., the reward signal is *sparse*, which makes the learning problem more difficult. The design of UPSIDE enables to identify the closest skill to the goal according to the learned discriminator, and we then fine-tune its diffusing part into a goal-oriented policy, as shown in Alg. 4.

The same approach is used for DIAYN and SMM. For SAC, a plain policy is trained directly on the reward signal.

We thus see that this task calls for a dual property of coverage and directedness.

Goals $g$ were sampled uniformly in the available state-space, but for the sake of simplicity, we only show in Section 5 two representative goal positions, a moderately close goal and a far goal. The goal region is a circle with radius 1, thus the agent gets rewarded 1, when $\|s - g\|_2^2 < 1$.

---

**Algorithm 3:** Unknown goal

**Input:** Unknown goal region $\mathcal{G}$, Budget $K$.

**for** $K$ iterations **do** // Find $\mathcal{G}$
  Sample $z$ in $\mathcal{T}$ at random
  Extract the sequence of nodes $z_{(1)}, \ldots, z$ in $\mathcal{T}$ leading to $z$
  Execute the composed (directed part) policy $(\pi_{z_{(1)}}, \ldots, \pi_z)$ followed by the diffusing part of $z$
  Stop if $\mathcal{G}$ is reached.
**if** $\mathcal{G}$ was found **then**
  Run Alg. 4 with goal $\mathcal{G}$
**else**
  Train SAC policy on the reward

---

**Algorithm 4:** Known goal

**Input:** Known goal region $\mathcal{G}$.
Compute skill-node

$$z^* = \arg\max_{z \in \mathcal{T}} \sum_{g \in \mathcal{G}} q_\phi(z|g).$$

Fine-tune the diffusing part of skill-node $z^*$ via RL with reward $r_\mathcal{G}(s) = \mathbb{1}[s \in \mathcal{G}]$.

---

604

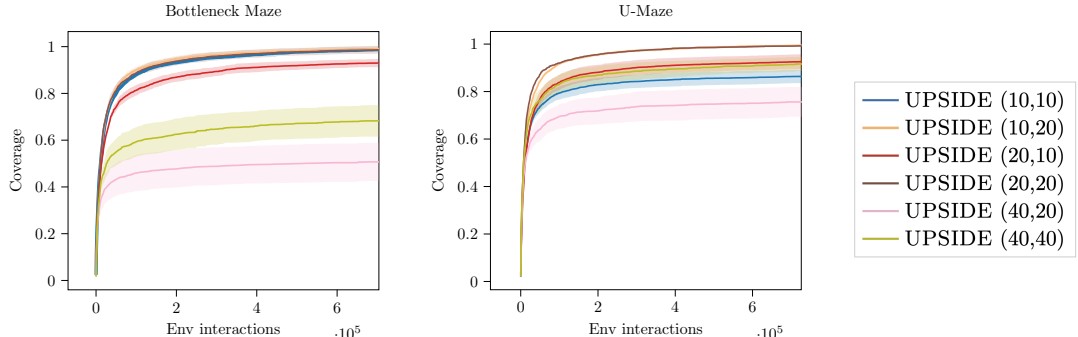

Figure 8: Ablation on the values of $T$ and $H$ for `UPSIDE` on the bottleneck and U-maze.

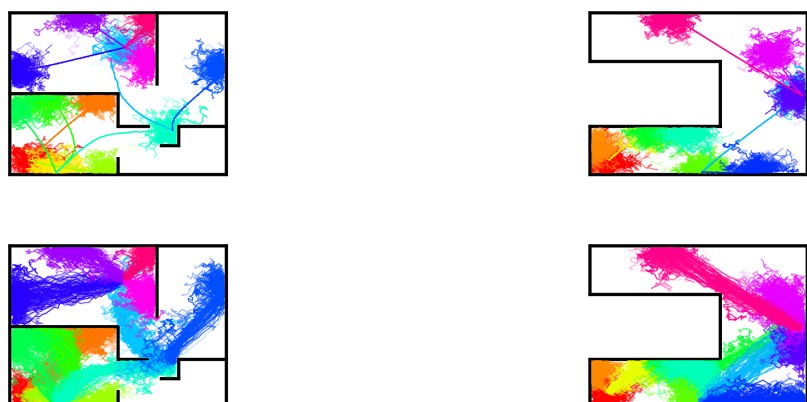

Figure 9: Difficulty of `UPSIDE` to cover the mazes if the hyperparameters $T, H$ are set too large w.r.t. the environment size (here, $T = H = 40$, and we recall that the mazes are of size $50 \times 50$). Top (resp. bottom) row corresponds to the stochastic (resp. deterministic) executions of the policies of the directed parts of the skills.

## D  Additional Experiments

In this section, we report additional experiments. We ran all methods with 3 unsupervised seeds for each set of hyperparameters. All plots are generated according to the evaluation protocol explained in App. C.4.

### D.1  Ablation on the skill lengths $T$ and $H$

We investigate the sensitiveness of `UPSIDE` w.r.t. $T$ and $H$, the lengths of the directed and diffusing part of the skill, respectively. Fig. 8 shows that the method is quite robust to reasonable choices of $T$ and $H$, although there exists configurations where `UPSIDE` does not achieve full coverage, in particular in the bottleneck maze when $T$ and $H$ are too large (e.g., $T = 40, H = 20$), see also Fig. 9. This makes sense as the environments require "narrow" exploration (e.g., the bottleneck region that the agent must "escape" from is quite small), thus composing disproportionately long skills may hinder the coverage. Moreover, increasing $T$ and $H$ makes the RL training longer and more challenging (e.g., the reward is more delayed).

### D.2  Visual example how the tree learned by `UPSIDE` fits the environment

We investigate the adaptivity (w.r.t. the input branching factor) of the tree structure of `UPSIDE` and illustrate that it can properly fit the unknown environment. As demonstrated in 10, `UPSIDE` successfully covers a large part of the tree maze, which is quite hard to explore given its narrow corridors. Here $T = 5$ and $H = 10$, and the branching factor $N_0$ is set to 3. In the terminal region

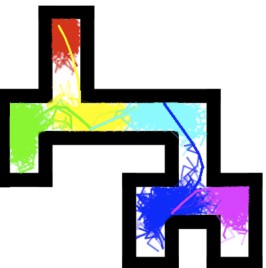 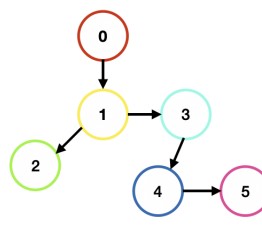

Figure 10: *(Left)* Unbalanced tree-shaped maze and *(Right)* the tree structure learned by UPSIDE. We see that it can successfully *map* the underlying structure of the unknown environment.

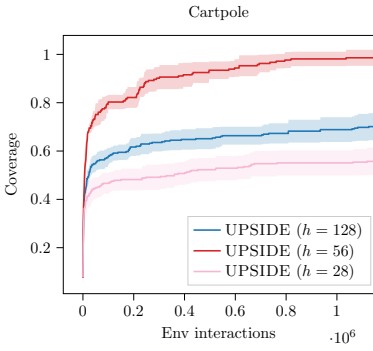

Figure 11: Ablation on $h$, the number of hidden units per layer of the discriminator of UPSIDE

of skill 1 (yellow), it is crucial to consolidate two skills 2 and 3 so that the tree can grow in both directions. While the tree may have expanded two skills 4 and 5 straight from 3, we see that the skill 4 (blue) overlaps with the intersection of the two small corridors, thus it is the only one sufficiently discriminable at this tree level, and UPSIDE covers the bottom right corridor in the subsequent level (i.e., from skill 4 to skill 5 in purple).

### D.3 Ablation on the capacity of the discriminator

On CartPole, we found that it was quite easy for the discriminator to separate skills, though they had close behaviors "visually". This can be explained by the fact that high-dimensional states are easier to discriminate. By reducing the capacity of the discriminator, skills would be naturally forced to be more "diverse" and avoid overfitting to certain state space regions. To verify this claim, we perform an ablation on the number of hidden units per layer of the discriminator (Fig. 11), which reveals that there is a sweet spot of hidden size where coverage is the best. When the hidden size $h$ is too big (128 or 256 in the main paper), many skills (more) are consolidated, but not diverse in their behavior, thus the coverage is not that large. On the other hand, when $h$ is too small, it is too hard to discriminate between skills.

### D.4 Results with more unsupervised seeds

In Fig. 12 we add results with 3 new unsupervised seeds per method and set of hyper-parameters. This complements Fig. 4 and 5 from the main paper by adding error bars. Compared to the main paper, training time was increased, thus explaining the slight differences in performance (e.g., UPSIDE, DIAYN-50, SMM-50 improve compared to the random policy thanks to training time).

### D.5 Average discriminator performance on Mazes and Cartpole

Fig. 13 reports the average discriminability of the skills (UPSIDE, DIAYN-curriculum and DIAYN-50) during training in Bottleneck maze, U-maze and Cartpole. We make the same observation as for Reacher (see rightmost plot of Fig 5). The DIAYN-50 skills (green) suffer from low discriminability,

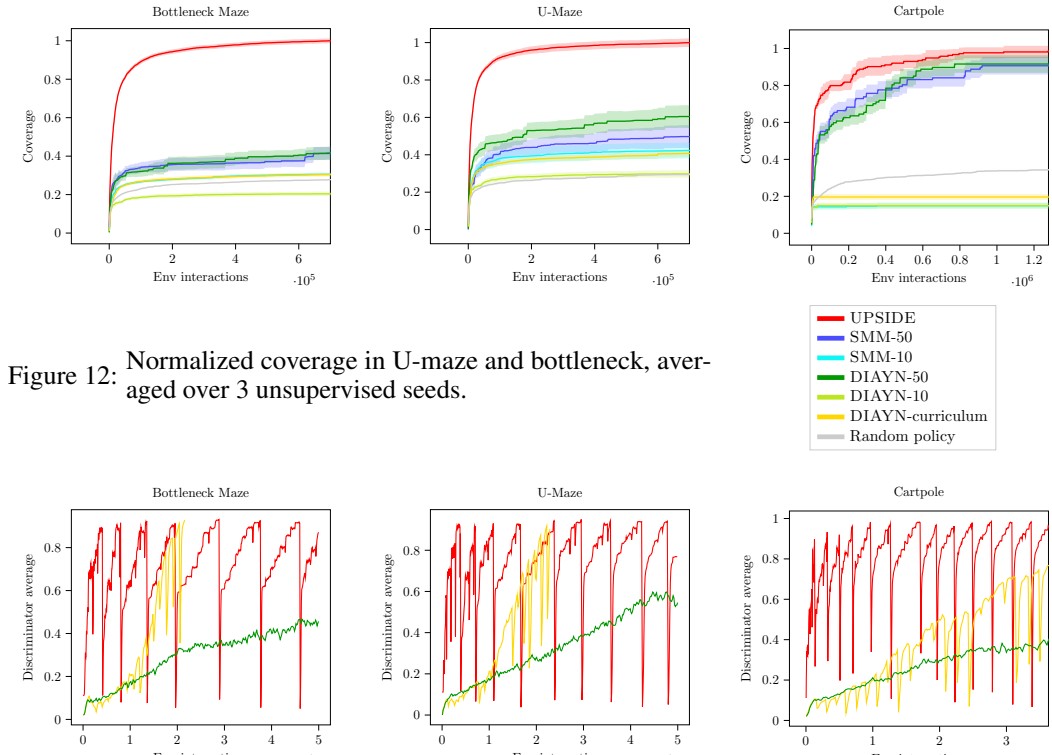

Figure 12: Normalized coverage in U-maze and bottleneck, averaged over 3 unsupervised seeds.

Figure 13: Average discriminability of the skills during training in Bottleneck maze, U-maze and Cartpole

while UPSIDE (red) (as well as DIAYN-curriculum in yellow) achieves much higher discriminability by removing redundant skills.

# E An interpretation of our optimization problem

In this section we provide a theoretically grounded interpretation of the optimization problem solved by UPSIDE in Sect. 3 and the extent to which it allows to tackle two downstream scenarios: known goal (Alg. 4) or unknown goal (Alg. 3). Throughout App. E, we consider that the MDP $M$ is finite-state, finite-action and communicating [35] (i.e., for every pair of states $(s, s')$, there exists a deterministic stationary policy under which $s'$ is accessible from $s$ in finite time with non-zero probability).

## E.1 Preliminaries

First we review some concepts and define notation. For any stationary policy $\pi$ and pair of states $(s, s')$, we denote by $\tau_\pi(s, s')$ the (possibly infinite) random variable of the hitting time of state $s'$ starting from state $s$ following policy $\pi$. We then define the *distance* $d_\pi$ as the expected hitting time of policy $\pi$, i.e.,

$$\tau_\pi(s, s') := \inf\{t \geq 0 : s_{t+1} = s' \mid s_1 = s, \pi\}, \qquad d_\pi(s, s') := \mathbb{E}[\tau_\pi(s, s')],$$

where the expectation is w.r.t. the random sequence of states generated by executing $\pi$ starting from state $s$. In addition, given a starting state $s_0$ and distance $d$, we define the *Max-Distance* as

$$D(d, M, s_0) := \max_{s \in \mathcal{S}} d(s_0, s).$$

We can instantiate two commonly considered distances.

- First, the *random-walk distance* is $d_{\text{RW}}(s, s') := d_{\pi_{\text{RW}}}(s, s')$, where $\pi_{\text{RW}}$ denotes a uniformly stochastic policy (i.e., whose executed actions are uniformly distributed, independently of the state of the MDP). Note that for $d \leftarrow d_{\text{RW}}$, $D$ corresponds to the *cover time* starting from $s_0$. This notion of complexity measures how hard it is, in expectation, to cover the entire state space of the MDP following a uniformly stochastic policy starting from $s_0$. It was studied in e.g., [26, 22, 11], leveraging graph theory [9].

- Second, the *shortest-path distance* is $d_{\text{SP}}(s, s') := \min_\pi d_\pi(s, s')$ (the minimum can be taken over the set of stationary deterministic policies [7]). Note that for $d \leftarrow d_{\text{SP}}$, $D$ corresponds to the *diameter* of the MDP [21, 40] and characterizes the complexity of navigating the state space starting from $s_0$ following the set of shortest-path policies. Note that for any state $s'$, $d_{\text{SP}}(\cdot, s')$ is the optimal value function under (undiscounted) reward function $r(s) = -\mathbb{1}[s \neq s']$. As such, numerous methods in goal-conditioned RL (explicitly or implicitly) target the set of policies that minimize $d_{\text{SP}}$ (or variants of it, such as discounted or horizon-truncated) [e.g., 2, 34].

## E.2 Interpretation

An interpretation of our approach is that it performs **clustering** over the state space based on **two different distance functions**:

- *A tight and difficult-to-deploy distance* $d_\star$. The tightest metric possible is to consider $d_\star \leftarrow d_{\text{SP}}$, the shortest-path distance.
- *A coarse and cheap-to-deploy distance* $d_+$. For example we can consider $d_+ \leftarrow d_{\text{RW}}$, the random-walk distance.

Specifically, our approach can be interpreted as seeking to minimize the intra-cluster distance $d_+$ while maximizing the inter-cluster distance $d_\star$. Although $d_+$ is coarser than $d_\star$, it had the advantage of being easier to execute the policy to which it corresponds (e.g., a random policy). The implicit assumption that we make is that $d_+$ is a *decent enough proxy* for $d_\star$ for *small horizon*, although it degrades sharply as the horizon increases.

In App. E.3 we analyze a simplified structure of our algorithm UPSIDE which allows us to theoretically analyze the extent to which UPSIDE can tackle the two downstream scenarios explained in App. C.5, depending on the environment's properties. For simplicity we will consider the "flat case" of UPSIDE (see Rmk. 2 for a discussion on the extension to the tree case). Before analyzing the downstream scenarios (App. E.4.1), we begin by analyzing the properties of the directed part (App. E.3.1) and the diffusing part (App. E.4) of each UPSIDE skill.

### E.3 An analysis of two downstream scenarios tackled by `UPSIDE`

#### E.3.1 Directed part of each `UPSIDE` skill

**Structure/Assumptions.**

- The directed part of each skill $k$ is characterized by a pair $(\pi_k, c_k)$ where the policy $\pi_k$ is of length $T$ and aims to attain a goal state $c_k \in \mathcal{S}$ (chosen by the skill).
- From the optimization problem of `UPSIDE`, each skill is $\eta$-consolidated according to the discriminator. We consider that the latter discriminates between the goals $\{c_k\}_{k \in [K]}$ given the current state. We then have that $\min_{k \in [K]} q_\phi(c_k|s_T) \geq \eta$.
- Finally, we assume that the predictions of the discriminator can serve as $\varepsilon_{\text{discr}}$-accurate approximations of the probability of $\pi_k$ reaching the centroid $c_k$ within its length of $T$ steps, where $0 \leq \varepsilon_{\text{discr}} < \eta$. This implicitly assumes that we can connect the discriminability property and the directedness property (respectively appearing in the reverse and forward forms of MI).

We first notice that the directed part $\pi_k$ has an intrinsic reward signal that approximately targets a goal-directed behavior. Indeed, as argued in [34, App. E], having an intrinsic reward signal of $r_z(s)$ scaling as $p(c_k|s)$ would amount to learning a goal-oriented policy with goal $c_k$. In particular, the optimal non-episodic policy $\pi^\dagger$ that minimizes $\mathbb{E}\left[\sum_{t=1}^{+\infty}(1 + \beta\mathcal{H}(\pi(\cdot|s_t)))\mathbb{1}[s_t \neq c_k]\right]$ induces a distance-to-goal of $d_{\pi^\dagger}(s, c_k) \leq (1 + \beta \log A)d_{\text{SP}}(s, c_k)$, i.e., it targets the shortest path up to an entropy bias. However, algorithmically, the directed parts are of length $T$ and the `UPSIDE` skills reset every $T + H$ steps. This episodic nature introduces a bias w.r.t. the optimal shortest-path behavior that is non-trivial to analyze and bound.

We now show that thanks to the constraint in the optimization problem of `UPSIDE` and by our assumption on the connection between the discriminability property and the directeness property, we can recover goal-directed properties for each first part of skills output by `UPSIDE`.

**Lemma 1.** *Any pair $(\pi_k, c_k)$ output by* `UPSIDE` *verifies*

$$d_{\pi_k}(s_0, c_k) \leq \frac{T + H + 1 - \eta + \varepsilon_{\text{discr}}}{\eta - \varepsilon_{\text{discr}}}.$$

*Proof.* Recall that the skill $k$ is episodic of length $T + H$, i.e., it resets to $s_0$ every $T + H$ time steps. We denote by $d_\pi^{(T+H)}$ the total number of steps before reaching either the skill's centroid or $T + H$ steps, and by $f_\pi^{(T+H)}$ the probability of failure to reach the centroid within $T + H$ steps. Then

$$d_{\pi_k}(s_0, c_k) \overset{(i)}{=} \frac{d_{\pi_k}^{(T+H)}(s_0, c_k) + f_{\pi_k}^{(T+H)}(s_0, c_k)}{1 - f_{\pi_k}^{(T+H)}(s_0, c_k)} \overset{(ii)}{\leq} \frac{T + H + f_{\pi_k}^{(T)}(s_0, c_k)}{1 - f_{\pi_k}^{(T)}(s_0, c_k)},$$

where (i) comes from [25, App. B.3], (ii) uses that $d^{(T+H)} \in [0, T + H]$ and that $f_{\pi_k}^{(T+H)}(s_0, c_k) \leq f_{\pi_k}^{(T)}(s_0, c_k)$. We now approximate the probability of failure of reaching the centroid by using the predictions of the discriminator (the more expressive the discriminator, the better the approximation): $|1 - f_{\pi_k}^{(T)}(s_0, c_k) - q_\phi(c_k|s_T)| \leq \varepsilon_{\text{discr}}$. The constraint of our optimization problem ensures that the pair $(\pi_k, c_k)$ output by `UPSIDE` satisfies $q_\phi(c_k|s_T) \geq \eta$. Therefore, it holds that

$$\frac{T + H + f_{\pi_k}^{(T)}(s_0, c_k)}{1 - f_{\pi_k}^{(T)}(s_0, c_k)} \leq \frac{T + H + 1 - q_\phi(c_k|s_T) + \varepsilon_{\text{discr}}}{q_\phi(c_k|s_T) - \varepsilon_{\text{discr}}} \leq \frac{T + H + 1 - \eta + \varepsilon_{\text{discr}}}{\eta - \varepsilon_{\text{discr}}}. \quad (6)$$

$\square$

Note that given any goal state $g$, having a policy $\pi$ with bounded $d_\pi(\cdot, g)$ is non-trivial, since it implies that it reaches the goal with probability 1 (i.e., that it is proper [7]). Also note that the "worst-case" discriminability property in the constraint (i.e., $q_\phi(c_k|s_T) \geq \eta$) is crucial to obtain Lem. 1, since it may not be possible to guarantee it given a discriminability property verified on average (e.g., via a conditional entropy term in the MI).

 ### E.4 Diffusing part of each UPSIDE skill

**733 Structure/Assumptions.**

734 • The diffusing part of skill $k$ is of length $H$ and is composed of a set of states radiating around
735 $c_k$, which acts as a centroid for the cluster of states generated by the diffusing part. Formally, we
736 consider that there exists $\delta > 0$ such that

$$\mathtt{DIFF}(k) := \big\{ y_k : \mathbb{P}(\tau_+(c_k, y_k) \leq H) \geq \delta \big\}, \tag{7}$$

737 where $\tau_+(s, s')$ denotes the hitting time following the policy that minimizes $d_+(s, s')$.

738 • According to the optimization problem solved by UPSIDE, the *clusters* associated to the $K$ skills
739 *saturate* the state space, i.e., we cannot consolidate an additional cluster. We propose to write this
740 condition as

$$\forall s \in \mathcal{S}, \ \exists k \in [K], \ \exists y_k \in \mathtt{DIFF}(k), \ \mathbb{P}\big(\tau_+(s, y_k) \leq H\big) \geq \delta, \tag{8}$$

741 otherwise from (7) it would be possible to consolidate an additional cluster with centroid $s$.

742 • Finally, we spell out an assumption on the environment that we make throughout App. E.4:

**743 Assumption 1.** *There exists $\Theta \geq 0$ such that*

$$\forall (s, s'), \quad \mathbb{P}(\tau_+(s, s') \leq H) \geq \delta \implies d_\star(s, s') \leq H + \Theta.$$

744 *This formalizes the assumption commonly made in goal-conditioned deep RL — either implicitly or*
745 *explicitly [e.g., 14, Sect. 3.3] — that if a goal is reachable, then there exists a policy that does so*
746 *reliably. Note that in the special case of a deterministic MDP we have $\Theta = 0$.*

**747 Definition 1.** *We define the following "local" quantities:*

748 • *For any $s \in \mathcal{S}$ and any $k \in [K]$, define $\Delta_+(s; k) := \max_{y_k \in \mathtt{DIFF}(k)} |d_+(s, y_k) - d_+(y_k, s)|$.*

749 • *For any $s \in \mathcal{S}$ and any $k \in [K]$, define $\Delta_\star(s; k) := \max_{y_k \in \mathtt{DIFF}(k)} |d_\star(s, y_k) - d_\star(y_k, s)|$.*

750 *Note that under the communicating MDP assumption, both quantities are always bounded. They*
751 *measure the level of "reversibility" of the MDP w.r.t. the $d_+$ and $d_\star$ distance, respectively. Moreover,*
752 *in the special case of an MDP with locally symmetric actions, the distance $d_\star$ is symmetric so $\Delta_\star = 0$.*

753 We first derive two lemmas and then position their statements w.r.t. the two downstream objectives of
754 UPSIDE.

**755 Lemma 2.** *It holds that*

$$\forall s \in \mathcal{S}, \ \exists k \in [K]: \quad \mathbb{P}\Big(\tau_+(c_k, s) \leq H + \frac{H + \Delta_+(s; k) + \Theta}{1 - \delta}\Big) \geq \delta^2.$$

**756 Lemma 3.** *It holds that*

$$\forall s \in \mathcal{S}, \ \exists k \in [K]: \quad d_\star(c_k, s) \leq 2(H + \Theta) + \Delta_\star(s; k),$$

757 *Proof of Lem. 2.* We prove the result by contradiction. Assume the contrary of Lem. 2; then there
758 exists a state $s \in \mathcal{S}$ such that for every $k \in [K]$, $\mathbb{P}(\tau_+(c_k, s) \leq H + Z_k) < \delta^2$, with $Z_k :=$
759 $(H + \Delta_+(s; k) + \Theta)/(1 - \delta)$. We now use that the diffusing part of each skill $k$ radiating around its
760 centroid $c_k$ is composed of states $\{y_k : \mathbb{P}(\tau_+(c_k, y_k) \leq H) \geq \delta\}$. This means that for every $k \in [K]$
761 and $y_k \in \mathtt{DIFF}(k)$, $\mathbb{P}(\tau_+(y_k, s) \leq Z_k) < \delta$. Noticing that $d_+(y_k, s) = \mathbb{E}[\tau_+(y_k, s)]$ by definition,
762 we get $d_+(y_k, s) > (1 - \delta)Z_k \geq H + \Theta + d_+(y_k, s) - d_+(s, y_k)$, where the last inequality comes
763 from the definition of $\Delta_+(s; k)$. Therefore, $d_+(s, y_k) > H + \Theta$. So by contraposition of Asm. 1,
764 $\mathbb{P}(\tau_+(s, y_k) \leq H) < \delta$. Since this is true for all $k \in [K]$ and $y_k \in \mathtt{DIFF}(k)$, we get a contradiction
765 on condition (8). □

766 *Proof of Lem. 3.* Take any state $s \in \mathcal{S}$. Case 1: $\exists k \in [K], s \in \mathtt{DIFF}(k)$. Then $\mathbb{P}(\tau_+(c_k, s) \leq$
767 $H) \geq \delta$. From Asm. 1 this means $d_\star(c_k, s) \leq H + \Theta$. Case 2: $\forall k \in [K], s \notin \mathtt{DIFF}(k)$. Then
768 from condition (8), there exists $k \in [K]$ and $y_k \in \mathtt{DIFF}(k)$ such that $\mathbb{P}\big(\tau_+(s, y_k) \leq H\big) \geq \delta$,
769 which implies that $d_\star(s, y_k) \leq H + \Theta$ from Asm. 1. By definition of $\Delta_\star(s; k)$, it holds that
770 $d_\star(s_k, y) \leq H + \Theta + \Delta_\star(s; k)$. Furthermore, $y_k$ verifies $\mathbb{P}(\tau_+(c_k, y_k) \leq H) \geq \delta$, which means
771 from Asm. 1 that $d_\star(c_k, y_k) \leq H + \Theta$. We conclude by the triangle inequality that $d_\star(c_k, y) \leq$
772 $d_\star(c_k, s_k) + d_\star(s_k, y) \leq 2(H + \Theta) + \Delta_\star(s; k)$. □

 **E.4.1 Analysis of two downstream scenarios tackled by** `UPSIDE`

774 We consider the two downstream tasks detailed in App. C.5: ① finding an unknown goal (Alg. 3) and
775 ② reliably reaching a known goal (Alg. 4).

776 These downstream scenarios require the ability to efficiently traverse from $s_0$ to any state $s$ of the
777 MDP. Ideally we would deploy the policy associated to $d_\star(s_0, s)$, i.e., the shortest-path policy, yet it
778 is difficult to compute. On the other extreme, deploying the random-walk strategy is very easy yet
779 much more inefficient, since $d_\star(s_0, s) \ll d_+(s_0, s)$. Our approach targets the following intermediate
780 approach.

781 First, we upper bound using the triangle inequality

$$\max_{s \in \mathcal{S}} d_\star(s_0, s) \ \leq \ \max_{s \in \mathcal{S}} \left\{ \min_{k \in [K]} d_\star(s_0, c_k) + d_\star(c_k, s) \right\}. \tag{9}$$

782 Under a **zero-shot downstream set-up**, the training objective of `UPSIDE` seeks to control the follow-
783 ing upper bound of (9)

$$\max_{s \in \mathcal{S}} \left\{ \min_{k \in [K]} d_\star(s_0, c_k) + d_+(c_k, s) \right\}. \tag{10}$$

784 Under a **few-shot downstream set-up**, `UPSIDE` fine-tunes the diffusing part of the skill to reach the
785 desired goal state. As such, it targets (9).

786 We now distinguish between the two types of downstream scenarios.

787 ① **The unknown-goal downstream task.**

788 From Lem. 2, whatever the unknown goal state $s$, there exists a skill $k$ whose diffusing part starting
789 from its centroid $c_k$ can reach $s$ with strictly positive probability, as long as it is executed long enough
790 (with length depending in particular on the *local* quantity $\Delta_+(s; k)$).

791 As such, Lem. 1 and Lem. 2 prescribe the following *algorithmic strategy*: in a round-robin fashion
792 over $k \in [K]$, execute the directed part of skill $k$ plus its diffusing part for *increasing* lengths (i.e.,
793 starting from $H$ and then gradually increasing it). The unknown goal should then be discovered at
794 some point, specifically within

$$O\left( \frac{1}{\delta^2} \left( \frac{T + H + 1 - \eta + \varepsilon_{\text{discr}}}{\eta - \varepsilon_{\text{discr}}} + \frac{H + \Delta_+(s; k) + \Theta}{1 - \delta} \right) \right)$$

795 time steps, by combining Lem. 1 and 2.

796 ② **The known-goal downstream task.**

797 From Lem. 3, for any known goal state $s \in \mathcal{S}$, there exists a skill $k \in [K]$ from which learning to
798 reach the goal $s$ can be facilitated. Indeed, the shortest-path distance from its centroid $c_k$ to the goal $s$
799 depends on the *local* quantity $\Delta_+(s; k)$ (as well as $H$ and $\Theta$).

800 As such, Lem. 1 (with (6)) and Lem. 3 prescribe the following *algorithmic strategy*: first reach the
801 centroid $c_k$ for which $k \in \arg\max q_\phi(c_k|s)$ (i.e., execute the directed part of skill $k$), and second
802 learn to reach $s$ from $c_k$ (by fine-tuning the diffusing part of skill $k$).

803 **Remark 1.** Inspecting the quantities in Def. 1 and in Asm. 1 allows to characterize the complexity
804 of the environment in tackling the two types of downstream tasks. In particular, we see that the
805 complexity is reduced in environments that are close to deterministic (i.e., smaller $\Theta$ in Asm. 1)
806 and that exhibit a "balanced / symmetric" behavior, with the least bottlenecks possible (i.e., smaller
807 quantities in Def. 1). In addition, the size of the state space $\mathcal{S}$ and the diameter of the MDP implicitly
808 play a role in the value of the number of clusters $K$ required and in the choice of $T$, which must be
809 large enough to ensure in Lem. 1 that $\eta > 0$ holds in the discriminator predictions.

810 **Remark 2.** In the tree case of `UPSIDE`, $T$ does not have to be large enough (as needed in the flat case)
811 since the state space may be covered by sequentially composing the directed parts of the skills of
812 length $T$. The equations from the flat case would look the same as in the flat case, yet two quantities
813 would be replaced: the probability of success of reaching the centroid of each cluster of skill-node $n$
814 would go from $\eta$ to $\eta^{d(n)}$ where $d(n)$ is the depth of skill-node $n$, and the length of the skill would
815 go from $T + H$ to $d(n)T + H$.