# OpenReview forum: "Direct then Diffuse: Incremental Unsupervised Skill Discovery for State Covering and Goal Reaching"
_NeurIPS.cc/2021/Conference — NeurIPS 2021 Submitted_

### Official Review · Reviewer_guNm · 2021-07-08

**Rating:** 6
**Confidence:** 3

**Summary:**

The paper presents a framework for an unsupervised learning of robotic skills that balances directness of exploration and coverage of state space. The theory of the method builds on the notion of mutual information between states and skills, where the conditional entropy between states and skill definitions corresponds to the directed part of the learned policy, and entropy of the states to the diffusing part of the policy. The directed part is optimized to increase discriminability of the skills given the states using a discriminator network. The diffusing part performs a random walk policy with uniform distribution over actions.The paper further proposes to optimize the number of skills by requiring the minimum discriminability of skills and discarding skills that fall below the discriminability threshold. Finally, in order to be able to reach states that are farther away, the paper proposes to combine skills in a tree structure and gradually expand the tree by adding new skill nodes to the tree. The approach is shown to outperform other unsupervised skill discovery methods on such tasks as continuous maze, CartPole and Reacher in OpenAI Gym during both, unsupervised skill discovery phase and training of downstream tasks.

**Limitations And Societal Impact:**

The authors discussed limitations of their work. The proposed approach does not lead to any obvious negative societal impacts.

**Main Review:**

Strengths
- Exploration and skill discovery is crucial for scaling up robot learning. Optimizing for both directness and exploration through mutual information between states and skills is an elegant way to acquire a wide variety of skills.
- The paper is well-written, and straightforward to understand and follow.
- The method is shown to discover a wider variety of skills and provide a better coverage of the state space than previous methods.

Criticism
- As discussed in the paper, and also visible in Fig. 3, the proposed method pushes the trajectories of different skills apart to increase their discriminability. Using a pre-defined minimum discriminability threshold might result in gaps in the state coverage, which can lead to low performance on tasks that require varying granularity of control, e.g. some manipulation tasks in robotics. It would be interesting to see a discussion how the method could deal with such tasks and ensure an appropriate state coverage.
- It would be useful to see more details about the skill latent variable z, in particular how it is constructed and its dimensionality.
- Experiments on downstream tasks assume that there is already a skill that is close to the new task and can be fine-tuned. It would be interesting to see a discussion how the method would deal with problems where combination of multiple skills is required.
- As also indicated in the paper, it would be interesting to see experiments on higher-dimensional problems.

**Time Spent Reviewing:**

3

---

> ### Author Response · Authors · 2021-08-11
> **Answer to reviewer guNm**
>
> Thank you for your valuable feedback. Please find our response to your comments below:
>
> **Minimum discriminability threshold:** The minimum discriminability threshold $\eta$ indeed plays an important role in the obtained coverage, as very strict discriminability, i.e. close to 1, would leave gaps in state coverage. To ensure the state-space is well-covered (i.e., no gaps) with some granularity of control, we can take a not-to-high discriminability threshold. In all our experiments, we chose $\eta = 0.8$ to authorize some overlapping between skills.
> Another parameter we can also play with is the length of the diffusing part: by increasing it at test time, we can eliminate some gaps in state-space coverage.
>
> **Skill latent $Z$:** We consider a discrete skill latent $Z=\{1, |Z|\}$, whose cardinality $|Z|$ is *adaptive* over time. Indeed, whenever a new skill-node is consolidated in the tree, we add $N_0$ new candidate skills (as its children nodes), and set $|Z| \leftarrow  |Z| + N_0$ (as explained in Alg. 2, Appendix A). Since UPSIDE adaptively adds or removes skills until “convergence”, $N_0$ can be equal to any value (i.e., either overspecified or underspecified w.r.t. to the effective space to consolidate skills in this region). This is particularly crucial as the maximum number of skills that can be fit in a region can vary a lot in an environment.
>
> In all our experiments we set $N_0=4$. While most prior approaches (e.g., VIC, DIAYN, SMM, EDL) rely on a preset number of skills $|Z|$, our method does not require such a hyperparameter: it creates a curriculum where $|Z|$ tends to increase over time, thus facilitating the learning, as also observed in VALOR.
>
> **Fine-tuning of skills for downstream task:**
> If the coverage of UPSIDE in the unsupervised phase is not good enough, a downstream task may indeed require solving a far-horizon problem (this problem is also true for DIAYN/SMM, or vanilla SAC). The UPSIDE tree would still be very useful, as it should enable the agent to reliably go from $s_0$ to the states reached by composition of skills, which can be much closer to the goal than $s_0$. As such, we may interpret the UPSIDE tree as “expanding” the narrow initial state distribution thanks to its paths of directed parts of skills.
>
> **High-dimensional problems:**
> Concerning exploration in high-dimensional space (e.g., images as observations), it is known to be a hard problem that is usually handled by projecting the high-dimensional states to a lower dimensional space (as it is done for instance in the Go-Explore algorithm). In UPSIDE, we expect to achieve reasonably well by using a discriminator with a strong inductive bias (CNN for instance on pixel-based observations) but we leave working on high-dimensional problems as a future work.

---

> > ### Author Response · Authors · 2021-08-25
> > **Has our response addressed your comments?**
> >
> > Hello reviewer guNm, we would be grateful if you can confirm whether our response has addressed your comments, and let us know if any issues remain.

---

> > > ### Comment · Reviewer_guNm · 2021-08-27
> > > **Thanks for your response**
> > >
> > > Thank you for clarifications. Given the concerns of other reviewers regarding variety of experiments, I would like to keep my rating.

---

### Official Review · Reviewer_1Gri · 2021-07-16

**Rating:** 6
**Confidence:** 4

**Summary:**

This method learns a space of intrinsic goals to maximize coverage of the state space. This work utilizes mutual information between skills and states, where the entropy of states and negative entropy of states given options describe the diffusing and directing objectives respectively. This is done by learning skills with two separate parts, a long scale directed component and a short scale diffusing component. These skills have state distributions that differ by at least a specified factor. It then separates these skills along a hierarchical tree structure, where temporal abstraction occurs by calling lower level options, which have a limited range, controlling the shape of the tree by pruning skills with low difference, and adding new skills when all the skills satisfy the condition.

**Limitations And Societal Impact:**

The work does not discuss limitations or societal impact. A discussion of how exploration could affect modern navigation methods, whether in maps, robot movement, etc, could have occurred.

**Main Review:**

Abstract: the abstract is somewhat confusing in that it is not clear whether the skills learns are diffusing or directed, or how that separation is made, since it is difficult for them to be both at once.

Introduction:
25 Unsupervised RL seems like an unusual phrase. Intrinsic rewards, exploration or unsupervised policy learning seem more descriptive. Reinforcement learning is defined by a reward function, distinct from supervised learning.

Figure 1 is fairly useful, but the description is somewhat verbose and between pictures it is not clear whether skills are being pruned or added, especially between image B and C / D and E.

42 Maximizing 1 does not encourage exploration in RL, though learning options which maximize 1 does tend to produce options which cover the space more fully.

58 From the introduction, it is still not clear how the tree structure is used and how it differs from the skill structure.

Setting:
It is not clear that a MDP is still an MDP without reward. This would just be an environment that exhibits Markovian dynamics. This is just a semantic issue, though.

Equation 1: While both forward and backward MI are mathematically equivalent, it is not clear from this section which one is more relevant for an intuition of this task. It seems like the first would be, however, since it better describes the diffuse and directed properties.

Algorithm Structure:
96,97,99 Note that the use of H(S), -H(S|Z) describe the forward form of mutual information, however p(z|s) is then used to describe the discriminability of a skill, but is from the reverse description of mutual information. This switching can be confusing.

Equation 2: While it makes mathematical sense that the minimum skill is less than the average, it is not entirely clear what a minimum level of discriminability is. In this case, it seems like intention of this lower bounding is to say that the loss is based on how discriminable the least distinct skills is.

Equation 3: It is also not clear how Equation 2 leads to Equation 3. While it makes sense to optimize over the number of skills, and complete the mutual information term, this is a consequence only after combining with Equation 1. It would be useful then to specify this.

135 This use of "convenient formulation" is unclear. This is used because maximizing over a cardinality of skills is an integer program, and thus using a greedy algorithm is the classic solution to such a problem.

In the final case, the skills are simply learned such that they exhibit better than \log \eta entropy over the states that they cover. It seems appropriate to relate this to clustering because this seems to have parallels to a kind of minimum distance coverage metric that might be used in that case.

3.3 It seems like the tree structure and how the composition works is integral to this work, but is only discussed in a limited manner. How nodes are chosen to be expanded upon and how an agent would navigate the tree seems like immediate questions.

3.4 While prior work has described how the discriminator q_\phi(z|s) is optimized, it is not entirely clear when reading this section what feedback is used to train the discriminator. This makes the description of the different steps confusing, since they simply assume that the discriminator reward performs well.

The pruning and growth method for skills in the tree is mentioned only briefly, and seems as though it requires a good number of hyperparameters to manage.

Related work:
From a formatting perspective, it would be best to separate the related work by category, especially since this work is primarily only interested in comparison with the line of work doing skill discovery based on mutual information. This is because this work tests only on the navigation domain used to visualize coverage, and uses the same language as these related methods.

Experiments:
While this method appears to perform substantially better than the existing methods, one weakness is that is does not compare with other methods which tend to escape some of the bottlenecks. In particular, though Explore, Discover and Learn: Unsupervised Discovery of State-Covering Skills is mentioned, and shown to get much better coverage by greater directedness, it is not compared against.

While the illustrations in Figure 3 are promising, it is worth noting that the performance of such exploration on high dimensional tasks has a much different characteristic. In particular, high dimensional states will take much longer to perform the dense coverage exhibited by the UPSIDE algorithm. This seems to limit the scope of results given (though most methods in this domain appear to struggle with this as well).

The diffusing component is not always tested in this method. For example, in the case where there would be a narrow bottleneck, it seems like the diffusing criteria could cause a skill that enters the bottleneck to fail because it cannot diffuse due to constraints in the environment. None of the experimental cases appear to test this possibility, but in many practical applications there are situations where this property is at play.

One ablative that seems important based on the results is to observe the level of effect of composing options. Since it seems like many of the other methods fail to escape the bottlenecks near the start, it seems like this difference is actually the primary advantage by comparison.

Overall, this work proposes a new and interesting way to define a set of skills that maximizes coverage over a space while keeping the skills distinct. The work build a novel framework of option composition and provides an alternative way to choose options---based on the worst differentiable option, which is interesting. Though it compares against methods which do not perform as well as other in the same domain, it does demonstrate encouraging results. However, the domain limits the scope of these results from many practical applications. Some missing components in the description and the equations does obfuscate exactly what the algorithm is doing.

Originality: decent

Quality: good

Clarity: marginal

Significance: marginal

**Time Spent Reviewing:**

3

---

> ### Author Response · Authors · 2021-08-11
> **Answer to reviewer 1Gri**
>
> Thank you for your valuable feedback, and for the suggestions to improve the presentation of the paper which we will incorporate in the revised version. Please find our response to your comments below:
>
>
> **Tree expansion:** A skill-node is chosen to be expanded upon if it meets the minimum discriminability threshold $\eta$ (see constraint in Eq. 5, Lines 14-21 in Alg. 2). The pruning and growth method for skills requires only two hyperparameters: $\eta$ as well as the patience $K$, which is the duration of the training of children skills before re-computing their empirical estimate $\widehat q_{\phi}^{(B)}(z)$ and thus re-evaluating whether they should be expanded upon or pruned.
> Since UPSIDE adaptively adds or removes skills until “convergence”, it is robust to the branching factor $N_0$ and it successfully adapts the number of skills that can be consolidated in a region. Notice that this property is crucial since the “effective” number of skills that can be consolidated may vary a lot in different regions of the environment, and thus at different nodes.
>
> **EDL baseline:** First, we would like to recall that EDL relies on the access to a distribution over valid states p(s). Availability of such an oracle is a strong prior knowledge requirement that considerably simplifies the exploration problem. In absence of this oracle, EDL needs to learn $p(s)$ following another exploration strategy, e.g. SMM. UPSIDE does not need such a distribution since it learns to explore directly from a single initial state.
>
> Following the submission deadline, we have implemented the EDL baseline on our environments. Although our bottleneck has a layout taken from EDL, note that it is 5x larger. We have observed that our implementation of EDL is unable to successfully escape the bottleneck and cover the entire state space (i.e., it performs similarly to DIAYN/SMM). Indeed, EDL relies on the success of SMM to obtain a good state distribution, which we already failed to make work in the bottleneck maze. Note that this is consistent with the EDL paper, whose Figure 1 relies on an oracle to sample states uniformly: “The maze with bottleneck states from Figure 1, where baseline approaches fail to explore a large extent of the state space, is a challenging environment where the limitations of EDL can be evaluated. We were unable to explore this type of maze effectively with SMM” (p.7). In contrast, UPSIDE is able to cover the entire maze *without resorting to any oracle*.
>
> **Ablations:** The two main components of UPSIDE that differ from e.g., DIAYN are the decoupled skill structure and the skill chaining via the growing tree. We have observed that both components are simultaneously required for good empirical performance. As suggested, we will include ablations illustrating this.
> As suggested by the reviewer, the tree structuring is key to improve exploration, as it enables the agent to learn on smaller and easier problems (with short-horizon MDPs) and thus mitigate the MI-based optimization issues (e.g., non-stationary rewards). A challenging aspect is to make skill composition work: we observed that our decoupling of skills into a directed -- low entropy -- and diffusing --high-entropy -- part, as a form of inductive bias on the entropy regularization, was critical. Indeed, without the diffusing part (only skill chaining), local coverage would not be guaranteed.
>
> *“It cannot diffuse due to constraints in the environment.”* -->  If the directed part of a skill ends just at the entrance of a narrow bottleneck, and its diffusing part does not escape the bottleneck, the directed part of some children skills will escape it to ensure discriminability.
>
> **High-dimensional problems:**
> Concerning exploration in high-dimensional space (e.g., images as observations), it is known to be a hard problem that is usually handled by projecting the high-dimensional states to a lower dimensional space (as it is done for instance in the Go-Explore algorithm). In UPSIDE, we expect to achieve reasonably well by using a discriminator with a strong inductive bias (CNN for instance on pixel-based observations) but we leave working on high-dimensional problems as a future work.

---

> > ### Author Response · Authors · 2021-08-25
> > **Has our response addressed your comments?**
> >
> > Hello reviewer 1Gri, we would be grateful if you can confirm whether our response has addressed your comments, and let us know if any issues remain.

---

> > > ### Comment · Reviewer_1Gri · 2021-08-30
> > > **Response to comments**
> > >
> > > Hello Authors, I was generally pleased with the response. I look forward to seeing the ablative results, especially to see differences with the EDL family of work.

---

### Official Review · Reviewer_HknL · 2021-07-17

**Rating:** 5
**Confidence:** 3

**Summary:**

The idea of this work is to maximize coverage while ensuring learned skills are distinguishable. The learning framework is based on maximizing mutual information (MI) between latent random variables and states.
This paper proposes a decoupled objective for local coverage and directness, on top of it, a tree structure based exploration method is proposed to incrementally compose learned skills for maximum coverage.

**Limitations And Societal Impact:**

The authors addressed the limitations and potential negative societal impact

**Main Review:**

While the problem studied in this work is important, there are several issues with clarity and novelty.

The authors need to define what is the length of “skill” before referring it.

The motivation of using the constrained optimization formulation for H(z) - H(z|s) (eq 3 and eq 4) should be elaborated more.

In introduction the objective seems to maximize H(s) - H(s|z) which aligns with its maximizing coverage motivation, but then in the method section, the objective becomes H(z) - H(z|s) which does not encourage explicit exploration, see EDL[Campos et al] and APS [Liu & Abbeel].

Also, EDL is a very related baseline, and should be discussed and compared in terms of methodology and experiments. EDL firstly optimize H(s) then use the data to optimize -H(s|z), which is closely related to this work. In fact, this work conducted experiments in 2D maze with the same topology as in EDL. I suggest the authors to compare their method against EDL.

------------------------

After reading the response from the authors, I am rising my score from 4 to 5.
As mentioned in the reply to the author response, while I tend to reject it, my score merely reflects the current status of the paper, with additional experiments in more convincing environments and downstream task fine-tuning, I think this work can be a strong paper.

**Time Spent Reviewing:**

2.5

---

> ### Author Response · Authors · 2021-08-11
> **Answer to reviewer HknL**
>
> Thank you for your valuable feedback. Please find our response to your comments below:
>
> **Skill length:**
> We thank the reviewer for raising this point. The maximum episode length, 200 in all our experiments (which is sufficient to reach all states in mazes), is the same for all methods. For the DIAYN/SMM, the skill length is set to the maximum episode length. On the other hand, in UPSIDE, the skill length is kept constant, e.g., $T=10$ (directed part) and $H=10$ (diffusing part), and the tree can be expanded at most up to a level $i_{\max} = \max \{ i : i T + H \leq \textrm{MaxEpisodeLength} \}$. We report an ablation of the sensitivity of $T$ and $H$ in App D.1.
> We believe this allows a fair comparison since every method (whether it chains skills like UPSIDE or not) is allowed the same maximum number of time steps between each environment reset.
>
> **Motivation for constrained optimization:** The motivation behind Eq. 3 (with the minimum instead of weighted average) and Eq. 4 (constrained formulation) is to enforce that *all* skills that are maintained by the algorithm are discriminable enough. Using a constrained formulation allows us to have explicit control over the discriminability of each skill, whereas the algorithmic approximations when optimizing the “regularized” formulation of Eq. 3 may alter the trade-off between its two summands. This is crucial for the tree expansion technique since it avoids consolidating and expanding “poor” skills, especially in lower levels of the tree, where it would lead to a significant waste of environment interactions and computational time. We will clarify this point in the revised version of the paper. Another way of seeing our constrained optimization is that instead of choosing a uniform $p(z)$ over all skills (as in DIAYN), we target a uniform $p(z)$ over only the discriminable-enough skills (and $p(z) = 0$ for the others).
>
> **EDL baseline:** First, we would like to recall that EDL relies on the access to a distribution over valid states p(s). Availability of such an oracle is a strong prior knowledge requirement that considerably simplifies the exploration problem. In absence of this oracle, EDL needs to learn $p(s)$ following another exploration strategy, e.g. SMM. UPSIDE does not need such a distribution since it learns to explore directly from a single initial state.
>
> Following the submission deadline, we have implemented the EDL baseline on our environments. Although our bottleneck has a layout taken from EDL, note that it is 5x larger. We have observed that our implementation of EDL is unable to successfully escape the bottleneck and cover the entire state space (i.e., it performs similarly to DIAYN/SMM). Indeed, EDL relies on the success of SMM to obtain a good state distribution, which we already failed to make work in the bottleneck maze. Note that this is consistent with the EDL paper, whose Figure 1 relies on an oracle to sample states uniformly: “The maze with bottleneck states from Figure 1, where baseline approaches fail to explore a large extent of the state space, is a challenging environment where the limitations of EDL can be evaluated. We were unable to explore this type of maze effectively with SMM” (p.7). In contrast, UPSIDE is able to cover the entire maze *without resorting to any oracle*.
>
> **On the two forms of MI:** Contrary to UPSIDE, EDL builds on the forward form of MI. While the latter explicitly captures the coverage/directedness trade-off, approximating it is difficult as hinted in the previous paragraph “EDL baseline”. Our method relies on the computationally more amenable reverse form of MI, which only needs to train a discriminator, i.e., learn a categorical distribution over the finite number of skills, instead of learning to predict possibly high-dimensional states from skills. The discriminator is a core component of our method, not only to train individual skills (as in DIAYN) but also to dictate the incremental growth of the tree. Therefore, we consider that relying on skill predictions via the discriminator to decide whether a skill has to be consolidated or not is easier than state predictions.
>
> As argued in prior work (EDL, APS), existing methods based on reverse MI (e.g., VIC, DIAYN, SMM) struggle to convincingly address the coverage/directedness trade-off, even in low-dimensional maze environments. Though UPSIDE doesn’t optimize directly for coverage and directedness,  the two key components of UPSIDE (decoupled skill structure as well as the chaining of the first parts of skills via a growing tree structure) inject inductive and structural biases about those two properties in the maximization of MI via the reverse form.

---

> > ### Comment · Reviewer_HknL · 2021-08-12
> > **thanks for the reply**
> >
> > Thanks for the effort in trying to compare with EDL, good to know the maze is 5x larger and more challenging than the maze used in prior work. It is actually not surprising to see that SMM failed to explore the maze effectively, which was shown in prior work Mutti et, al, and therefore EDL is unable to learn skills effectively.
> >
> > Appreciate the clarification on the motivation of constrained optimization and the comparison with prior mutual info based methods, I believe having these clarification in the updated version will greatly improve the readability.
> >
> > I will update my reviews and raise the rating to 5 in light of these changes.
> >
> > While the idea is novel, the experiments are not convincing enough though. Apart from maze and Cartpole, the method is only evaluated in Reacher, which arguably is one the most simplest or erroneous environments in mujoco. And as the authors mentioned, highly stochastic actions lead to excellent state coverage, making it not a suitable environment for showing the benefit from learning skills for exploration, and vice versa.
> > I would suggest the authors to evaluate UPSIDE in environments that require more skills discovery, like Hopper, Ant, and Walker. Humanoid might be too challenging but would be nice to have it as well since prior methods like DIAYN do not work at all in it, maybe UPSIDE is a promising method.
> >
> > In addition to skill learning, prior work like Mutti et al (figure 3) evaluate the performance of various methods including various skill methods including SMM and ICM in downstream task adaptation, I believe UPSIDE would benefit from evaluating in these downstream tasks.
> >
> > While I tend to reject it, my score merely reflects the current status of the paper, with additional experiments in more convincing environments and downstream task fine-tuning, I think this work can be a strong paper.
> >
> > Mutti, Mirco, Lorenzo Pratissoli, and Marcello Restelli. "Task-Agnostic Exploration via Policy Gradient of a Non-Parametric State Entropy Estimate." Proceedings of the AAAI Conference on Artificial Intelligence. Vol. 35. No. 10. 2021.

---

> > > ### Author Response · Authors · 2021-08-25
> > > **Response to Reviewer HknL**
> > >
> > > Thanks for the reply! While we agree that additional domains may provide further insights, we believe that the environments that we included in the current version are well suited to focus on the skill discovery aspect, without the complexity burden falling on the underlying RL algorithm, as argued in [Campos et al., 2020, Sect. 4].
> > >
> > > We point out that we also evaluate UPSIDE on downstream tasks (see Fig. 6), where the objective is to identify an unknown goal location (to evaluate UPSIDE’s coverage property) and then reliably reach it (to evaluate UPSIDE’s directedness property). This is similar to [Mutti et al., 2021, Fig. 3] although the goal location is known in their case. We show that UPSIDE outperforms SMM (see lines 302-306) and we decided to not include ICM because [Lee et al., 2019] already shows that it is outperformed by SMM.

---

### Official Review · Reviewer_tU7w · 2021-07-20

**Rating:** 6
**Confidence:** 5

**Summary:**

The authors propose a novel skill discovery method that combines standard mutual information maximization with notable changes. The first is that skill now enter a diffusion stage (random actions) before being predicted by a reverse predictor. The second is that environmental resets are exploited in order to construct a skill tree, wherein new skills are executed only after their parent skills. The final change is a pruning/growth strategy similar to VALOR, which is critical for constructing the skill tree.

The authors evaluate their method by comparing it to traditional skill discovery methods in terms of state coverage and performance on downstream tasks.

**Limitations And Societal Impact:**

Environmental resets are mentioned, but should also be addressed as a limitation. Other skill discovery methods can easily be extended non-episodic environments (e.g. compare DIAYN to VISR), but UPSIDE relies heavily on this ability in order to establish a root node for its tree.

**Main Review:**

Overall, I really like this idea as well as its execution. The tree structure and separate diffusion process tie together nicely (i.e. no diffusion until leaf node), and the qualitative results are very strong. But there are a few things stopping me from a stronger acceptance recommendation.

There are two critical ablations that are missing. The first is to run UPSIDE without the diffusion stage (likely requiring retuning the action entropy bonus) to see how necessary it truly is. The second is to run UPSIDE without the tree structuring. My hypothesis is that the diffusion stage is only necessary with the tree structure, as it prevents stochasticity from hurting the ability to reach leaf nodes, though perhaps just executing the deterministic policy with otherwise high action entropy bonus would be sufficient. In particular, I don't buy the claim on line 121 that direct-then-diffuse is qualitatively different from direct-with-action-entropy, since e.g. the policy could just be super stochastic after becoming predictable -- effectively replicating the direct-then-diffuse behavior while maintaining the ability to utilize other strategies (e.g. don't randomly take an action that kills you).

In addition, your baselines would be a lot more impactful if you swept over skill duration. UPSIDE can chain together skills, which allows for its impressive performance on e.g. unknown far goal, but perhaps DIAYN could also achieve this with sufficient skill duration.

One final point is that the claim on line 198 that VIC/DIAYN don't have an unsupervised model selection criteria is simply untrue. Its just the value of the lower-bound e.g. E[ log p(z) - log q(z | s) ]. Indeed exponentiating this even gives a value for the effective number of skills, which is more readily interpretable than UPSIDE's actual number of skills, since those won't all be perfectly interpretable. Of course, this criteria could easily be used by UPSIDE as well, and UPSIDE has enough other advantages to offset the loss of this contribution.



**Time Spent Reviewing:**

4

---

> ### Author Response · Authors · 2021-08-11
> **Answer to reviewer tU7w**
>
> Thank you for your valuable feedback. Please find our response to your comments below:
>
> **Ablations:** The two main components of UPSIDE that differ from e.g., DIAYN are the decoupled skill structure and the skill chaining via the growing tree. We have observed in our experiments that both components are simultaneously required for good empirical performance. As suggested, we will include ablations to illustrate this need. We now provide some insights on why this is the case:
>
> As interestingly mentioned by the reviewer, a direct-with-action-entropy strategy (like DIAYN) could, per se, recover this decoupled behavior and cover well. However, we have not observed this behavior consistently and the achieved performance was worse: this is due to the fact that learning on non-stationary rewards is already hard, but also trying to maximize entropy is even harder. Decomposing the skill into i) a directed -- low entropy -- part and ii) a diffusing -- high-entropy  -- part can be seen as a form of inductive bias that makes the learning easier.
> This decomposition is particularly useful when used in combination with the tree structure: the directed part allows chaining skills in a consistent way and it facilitates expanding exploration from the initial state to further regions of the environment. On the other hand, the diffusing part keeps providing the local exploration that is required to provide sufficient coverage. Without the skill structure (e.g., with a tree-based version of DIAYN), skill chaining for exploration and local coverage would not be as effective.
>
> **Skill duration of the baselines (e.g., DIAYN):**
> We thank the reviewer for raising this point. The maximum episode length, 200 in all our experiments (which is sufficient to reach all states in mazes), is the same for all methods. For the DIAYN/SMM, the skill length is set to the maximum episode length. On the other hand, in UPSIDE, the skill length is kept constant, e.g., $T=10$ (directed part) and $H=10$ (diffusing part), and the tree can be expanded at most up to a level $i_{\max} = \max \{ i : i T + H \leq \textrm{MaxEpisodeLength} \}$. We report an ablation of the sensitivity of $T$ and $H$ in App D.1.
> We believe this allows a fair comparison since every method (whether it chains skills like UPSIDE or not) is allowed the same maximum number of time steps between each environment reset.
>
> **Unsupervised model selection criteria**: The reviewer is right in pointing out that existing algorithms (e.g.,VIC/DIAYN) can compute an effective number of skills (via a data-driven lower bound on the MI) and also use it as an unsupervised model selection criterion. UPSIDE’s number of skills is actually tightly connected to this criterion, since the discriminability constraint ensures that each consolidated skill is sufficiently interpretable. For DIAYN and SMM, our model selection is based on the final discriminator average value, which is directly related to the effective number of skills due to the uniform $p(z)$ assumption. We will clarify this point in the final version of the paper.

---

> > ### Author Response · Authors · 2021-08-25
> > **Has our response addressed your comments?**
> >
> > Hello reviewer tU7w, we would be grateful if you can confirm whether our response has addressed your comments, and let us know if any issues remain.

---

> > > ### Comment · Reviewer_tU7w · 2021-08-25
> > > **Thank you for the clarifications.**
> > >
> > > Thank you for the clarifications.
> > >
> > > I'm glad you'll include the requested ablations, but any chance you could share these results during this discussion period? e.g. a results table and/or an annon link to qualitative results?
> > >
> > > Additionally, I'm afraid I agree with Reviewer HknL that the experiments lack sufficient variety. A few more control environments (e.g. half-cheetah, ant) would substantially increase my confidence in the generality of this method.

---

### Decision · Program_Chairs · 2021-09-27

**Decision:**

Reject

**Comment:**

Reviewers found the proposed method interesting and early results promising, but ultimately the current set of experiments feel unfinished and fall short of acceptance. First, only a few relatively simple environments are currently considered. Reviewers felt that it is important to see UPSIDE succeed in more challenging environments that have been used for skill learning, e.g. Hopper, Ant, or Walker in Mujoco. Second, two important ablations are missing - the removal of the diffusion stage, and the remove of the tree structuring of skills. While the authors claim both are important for their method, it is important to demonstrate this to readers. With these two (relatively simple) issues addressed, I expect UPSIDE will be a worthy contribution to the community.